# *Anopheles* mosquitoes reveal new principles of 3D genome organization in insects

Varvara Lukyanchikova[1,2,3,4,11], Miroslav Nuriddinov [3,4,11], Polina Belokopytova[3,4], Alena Taskina[3,4], Jiangtao Liang [1,2], Maarten J. M. F. Reijnders[5], Livio Ruzzante [5], Romain Feron [5], Robert M. Waterhouse [5], Yang Wu[2,6,7], Chunhong Mao[8], Zhijian Tu[2,6], Igor V. Sharakhov [1,2,9,12✉] & Veniamin Fishman [3,4,10,12✉]

Chromosomes are hierarchically folded within cell nuclei into territories, domains and subdomains, but the functional importance and evolutionary dynamics of these hierarchies are poorly defined. Here, we comprehensively profile genome organizations of five *Anopheles* mosquito species and show how different levels of chromatin architecture influence each other. Patterns observed on Hi-C maps are associated with known cytological structures, epigenetic profiles, and gene expression levels. Evolutionary analysis reveals conservation of chromatin architecture within synteny blocks for tens of millions of years and enrichment of synteny breakpoints in regions with increased genomic insulation. However, in-depth analysis shows a confounding effect of gene density on both insulation and distribution of synteny breakpoints, suggesting limited causal relationship between breakpoints and regions with increased genomic insulation. At the level of individual loci, we identify specific, extremely long-ranged looping interactions, conserved for ~100 million years. We demonstrate that the mechanisms underlying these looping contacts differ from previously described Polycomb-dependent interactions and clustering of active chromatin.

[1] Department of Entomology, Virginia Polytechnic Institute and State University, Blacksburg, VA 24061, USA. [2] Fralin Life Science Institute, Virginia Polytechnic Institute and State University, Blacksburg, VA 24061, USA. [3] Institute of Cytology and Genetics SB RAS, Novosibirsk, Russia. [4] Novosibirsk State University, Novosibirsk, Russia. [5] Department of Ecology and Evolution, University of Lausanne and Swiss Institute of Bioinformatics, 1015 Lausanne, Switzerland. [6] Department of Biochemistry, Virginia Polytechnic Institute and State University, Blacksburg, VA 24061, USA. [7] Department of Pathogen Biology, School of Public Health, Southern Medical University, 510515 Guangzhou, Guangdong, China. [8] Biocomplexity Institute & Initiative, University of Virginia, Charlottesville, VA 22911, USA. [9] Department of Genetics and Cell Biology, Tomsk State University, Tomsk, Russia. [10] AIRI, Moscow, Russia. [11]These authors contributed equally: Varvara Lukyanchikova, Miroslav Nuriddinov. [12]These authors jointly supervised this work: Igor V. Sharakhov, Veniamin Fishman. ✉email: igor@vt.edu; minja-f@ya.ru

Three-dimensional (3D) genome organization has recently been recognized as a complex and dynamic mechanism of gene regulation. Understanding of these features has been extensively advanced by the development of chromosome conformation capture (3C) methods, which enable genome-wide chromatin contacts to be explored at fine resolution[1–3]. In addition, data obtained using 3C-technologies help to generate high-quality chromosome-level genome assemblies, facilitating comprehensive analysis of genome evolution[4–7].

Comparative studies performed on multiple vertebrate species revealed that genome architecture is evolutionarily conserved and could be explained by the dynamic interplay between processes of cohesin-mediated loop extrusion and chromatin compartmentalization[8–11]. In insects, comprehensive analyses[12–15] and cross-species comparisons[5–7] of genome architecture have to date focused only on *Drosophila* species. These studies suggested that, in contrast to mammals, CTCF-mediated insulation plays only a limited role in the organization of chromatin contacts in *Drosophila* genome[16,17]. Instead, separation of active and repressed chromatin plays an essential role in the formation and interaction of topologically associated domains (TADs) or compartmental domains, which are basic units of chromatin organization in *Drosophila*[16,18].

Recently, a study using *Drosophila* lines with highly rearranged genomes[19] suggested that disrupting TADs does not influence coordinated gene expression. Consistent with this conclusion, Torosin et al. demonstrated that the majority of TADs have been reorganized since the common ancestor of *D. melanogaster* and *D. triauraria*, two species that diverged ~15 million years ago[6]. In contrast, Renschler et al. used three distantly related *Drosophila* species (~49 million years of divergence) to show that while chromosomal rearrangements might shuffle the positions of entire TADs, they are preferentially maintained as intact units[5]. In another study, Liao and co-authors compared *D. pseudoobscura* and *D. melanogaster*, which are separated by ~49 million years of divergence, and showed that ~30–40% of their genomes retain conserved TADs[7]. Thus, the roles of 3D chromatin interactions in the function and evolution of insect genomes remain unclear.

To resolve the apparently conflicting observations of TAD formation in mammals and fruit flies as well as of TAD evolution in drosophilids, investigations of chromatin organization in other groups of insects could be useful. To address this, we characterized the chromosomal-level genome architectures of five *Anopheles* mosquito species using a Hi-C approach. *Anopheles* mosquitoes are exclusive vectors of human malaria, which has a devastating global impact on public health and welfare. Comparative genomic analyses of multiple *Anopheles* species[20] revealed a high rate of chromosomal rearrangements, especially on the X chromosome, making mosquitoes an attractive model for studying interconnections between structural genome variations, chromosome evolution, and chromatin architecture.

Using Hi-C data we improved existing genome assemblies of three mosquito species and generated new chromosome-level assemblies for two others. Supplemented with RNA-seq and ChIP-seq profiles, these data allowed us to perform comprehensive characterization of epigenomes in malaria mosquitoes and dissect the principles underlying chromatin architecture. Interspecies comparisons showed little evidence of the link between chromatin organization and chromosomal evolution, although we found that the pattern of chromatin contacts remains remarkably stable within syntenic blocks.

## Results

**Hi-C data guided chromosome-level assembly of five *Anopheles* mosquito genomes.** In the Hi-C experiment, we used 15–18 h embryos of mosquito species from three different subgenera of the *Anopheles* genus: *Cellia* (*An. coluzzii*, *An. merus*, *An. stephensi*), *Anopheles* (*An. atroparvus*) and *Nyssorhynchus* (*An. albimanus*) (Fig. 1a–d). In addition, we obtained and sequenced Hi-C libraries from an adult *An. merus* mosquito. Phylogenetic relationships of the selected species represent a broad range of evolutionary distances, from 0.5 million years (MY) between the closely related species of the *An. gambiae* complex - *An. coluzzii* and *An. merus*[21], to 100 MY separating the most distant lineage of *An. albimanus* from the rest[20] (Fig. 1a).

After sequencing the libraries and merging biological replicas, we obtained 60–194 million unique alignable reads for each species. Library statistics show high quality of the obtained data (Supplementary Data 1).

Assembling *Anopheles* genomes to chromosomal levels has been challenging mainly due to the presence of highly repetitive DNA clusters, which regular Illumina sequencing and assembly could not successfully resolve[20]. When selecting species for Hi-C data generation, chromosome-length Illumina assemblies based on physical mapping were already available for *An. albimanus* and *An. atroparvus*[22,23]. A PacBio assembly for *An. coluzzii*, an Illumina assembly for *An. merus*, and a combined 454, PacBio, and Illumina assembly for *An. stephensi* were only at scaffold-level with N50s of 3.5 Mb[24], 2.7 Mb[20], and 1.6 Mb[25], respectively. While later evolutionary superscaffolding and chromosomal anchoring improved the *An. merus* and *An. stephensi* assemblies, they did not reach a full chromosomal level[26].

For *An. merus*, we performed PacBio sequencing using whole genomic DNA extracted from 100 adult males, which produced reads with an average read length N50 of 2.7 Mb (Supplementary Data 2). We employed a 3D-DNA pipeline[4] to de novo assemble the new PacBio *An. merus* scaffolds and existing *An. coluzzii*[24] and *An. stephensi* genomic scaffolds into chromosomes using our Hi-C datasets. We also re-scaffolded the available chromosomal Illumina assemblies of *An. albimanus*[22] and *An. atroparvus*[23] (Supplementary Table 1). Misassemblies and chromosome rearrangements were detected and the misassemblies were fixed manually. The available physical genome maps for these species were used to verify the corrections[22,23,27–29]. Applying Hi-C scaffolding to the existing AgamP4 *An. gambiae* PEST assembly[30,31] revealed multiple errors in the AgamP4 scaffolds. For this reason, instead of the *An. gambiae* genome, we scaffolded PacBio contigs obtained from another strain of *An. coluzzii* - Ngousso colony[24], which is the closest relative to *An. gambiae*[21]. The obtained *An. coluzzii* assembly was more accurate and more useful for further analyses than the Hi-C scaffolded AgamP4. Thus, Hi-C data allowed us to generate five new chromosome-level assemblies: AalbS4, AatrE4, AcolN2, AmerM5, and AsteI4 (see Supplementary Table 1 for N50 metrics and Supplementary Table 2 for BUSCO scores).

Pairwise alignments of the resulting assemblies showed that the lengths of alignment blocks and percentages of alignable nucleotides (ranges from 19 to 93%) correlated with evolutionary distances among species (Fig. 2A). For all the species, five large scaffolds corresponding to the chromosomal arms (X, 2R, 2L, 3R, and 3L) were identified. In species of the *An. gambiae* complex, *An. coluzzii* and *An. merus*, arms are denoted as chromosomal elements 1 (X), 2 + 3 (2R + 2L), 4 + 5 (3R + 3L). The correspondence of chromosomal arms in other species is as follows: *An. stephensi* 1, 2 + 5, 3 + 4; *An. atroparvus* 1, 4 + 3, 2 + 5; *An. albimanus* 1, 2 + 4, 5 + 3[20]. Therefore, *An. stephensi* and *An. atroparvus* have the same arm association, although they have different arm names for the same chromosomal elements.

Using the alignment results, we defined blocks of conserved synteny as well as synteny breakpoints for each pair of species (see "Methods" section). As expected, the average number of synteny blocks increases with the evolutionary distance, whereas their length tends to decrease (Fig. 2a, b). The vast majority

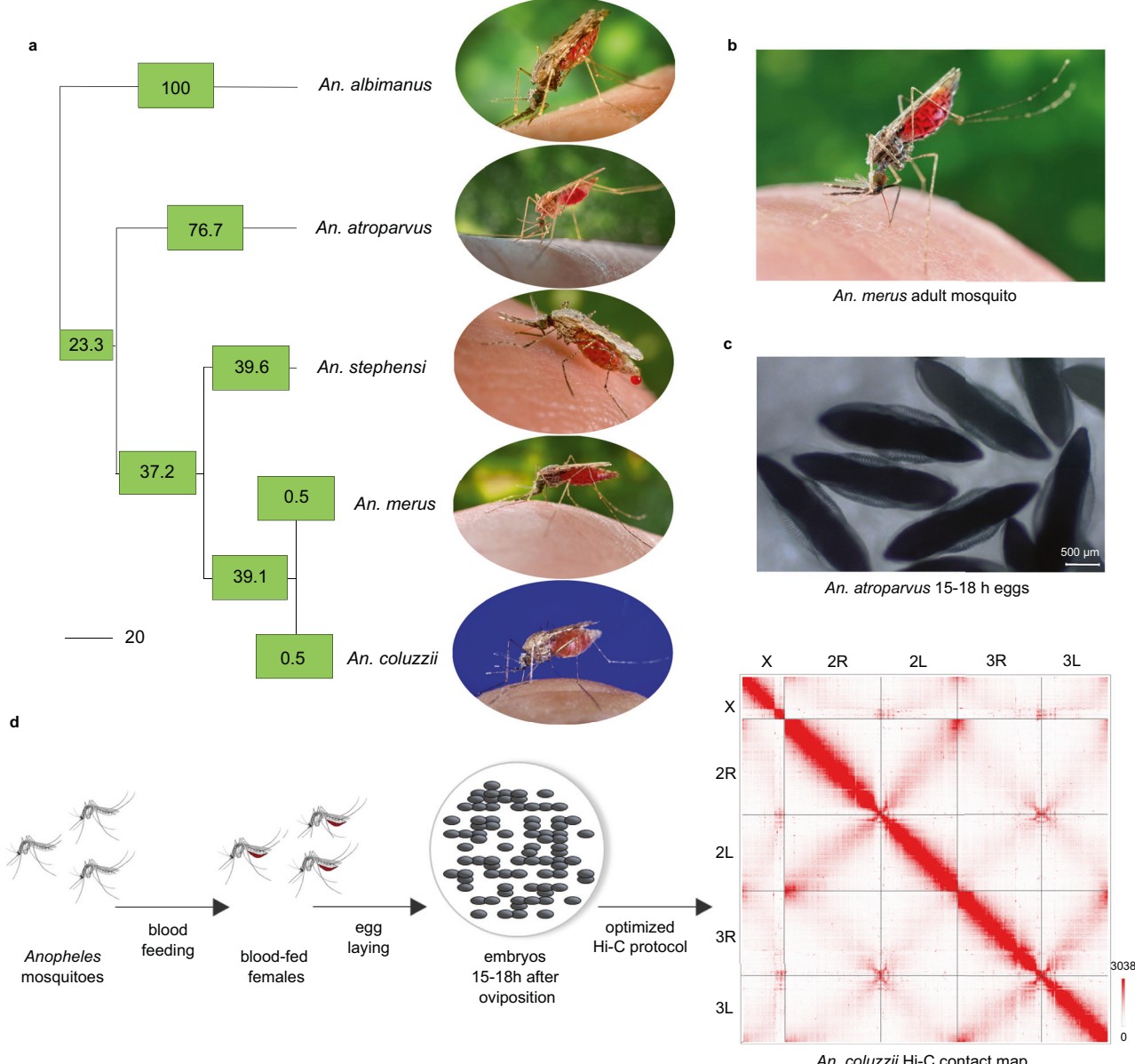

**Fig. 1 Selected species and Hi-C experimental setup. a** A time-calibrated phylogenetic tree shows estimated evolutionary distances among the selected *Anopheles* species; numbers in boxes show divergence times in millions of years (MY) for each branch; the scale bar represents 20 MY; **b** *An. merus* adult female mosquito; **c** *An. atroparvus* embryos at the 15–18 h developmental stage; **d** The experimental design of the Hi-C experiments using mosquito embryos, color bar reflects the contact frequency. Adult mosquito photo credit: CDC/James Gathany.

of rearrangements occur within the same chromosomal arm, and even for the most evolutionary distant species, inter-chromosomal translocations are rare (Fig. 2c and Supplementary Fig. 1A–F). Comparing individual chromosomes, we found that synteny blocks on the X chromosome were smaller than synteny blocks on autosomes for all the species (Fig. 2d). This finding is in agreement with the previously shown elevated gene shuffling on the X chromosome[20]. Overall, the Hi-C data allowed us to improve the existing genome assemblies for two *Anopheles* species (*An. albimanus* and *An. atroparvus*) and generate de novo chromosome-level assemblies for three species (*An. coluzzii, An. stephensi,* and *An. merus*).

**Hi-C data identifies polymorphic inversions.** Our chromosome-length Hi-C contact maps of *Anopheles* species identified a total of four inversions, ranges from 2.8 to 16 Mb in length

(Supplementary Table 3 and Supplementary Fig. 2). Inversion breakpoint regions are characterized by "butterfly" contact patterns, which were associated previously with balanced inversions[32]. We considered an inversion polymorphic if we did not see the complete disruption of interactions near the break-point loci on the main diagonal of the Hi-C map. The most prominent example of such contacts was found on the chromosomal arm 2R *An. stephensi* (Fig. 3a and Supplementary Fig. 2A), where the Hi-C map suggests an inversion of a large (~16 Mb) chromosomal region. We observed both off-diagonal long-range interactions, as well as interactions near the diagonal suggesting that both inverted and standard arrangements are present in the population. This paracentric inversion corresponds to the previously described polymorphic inversion 2Rb in *An. stephensi* populations[33–35]. The boundaries of this inversion on chromosome 2R are in agreement with our cytogenetic analysis of the same *An. stephensi* colony (Fig. 3b), and with previous cytological

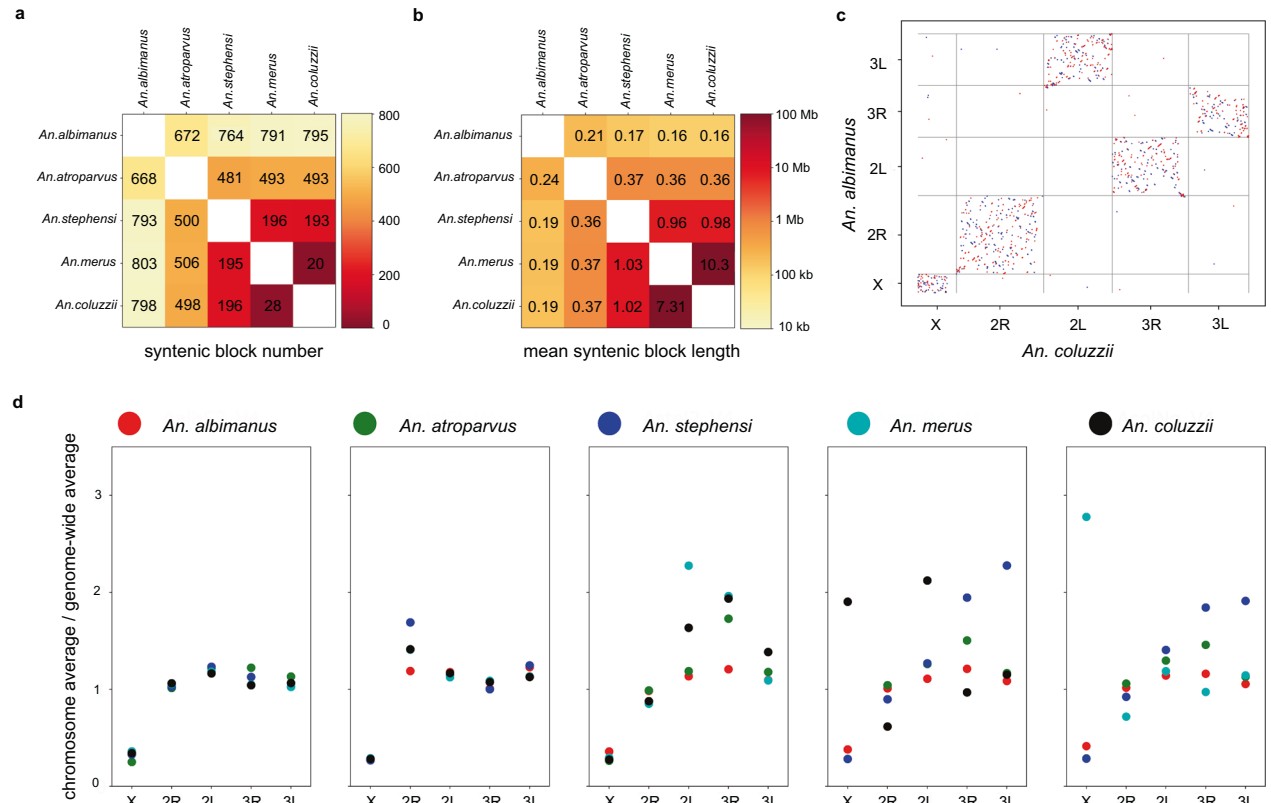

**Fig. 2 Comparison of alignability and rearrangements among five chromosomal-level *Anopheles* genomes. a** Heatmap representing the number of synteny blocks for each pair of genomes; **b** Heatmap representing the mean lengths of synteny blocks for each pair of genomes; **c** Synteny dot plot showing results of pairwise alignments between *An. coluzzii* and *An. albimanus*; forward orientation visualized in red, reversed in blue; **d** Comparison of syntenic block lengths for each chromosome arm. *Y*-axes represent the ratio of average syntenic block length on the specific chromosome (depicted on *X*-axes) to the genome-wide average. Titles of plots indicate alignment reference and colors of dots correspond to query species: red—*An. albimanus*, green—*An. atroparvus*, indigo—*An. stephensi*, azure—*An. merus*, black—*An. coluzzii*.

data for the Indian wild-type laboratory strain of *An. stephensi* (Fig. 3c)[25]. The dot plot alignment of our genome assembly with the recently PacBio-sequenced genome of the *An. stephensi* UCISS2018 strain showed that the AsteI4 assembly has a standard arrangement, while the assembly of the UCISS2018 strain has the inverted 2Rb arrangement[36]. Here we, for the first time, report the approximate genomic positions of breakpoint regions for the *An. stephensi* 2Rb inversion (Supplementary Table 3). Examination of the *An. atroparvus* Hi-C contact map allowed us to discover a previously unknown polymorphic inversion on the 2L arm (Fig. 3d and Supplementary Fig. 2, B). We confirmed the presence of this inversion on the preparations of polytene chromosomes from *An. atroparvus* and we named it 2L1 (Fig. 3e). The 2L1 inversion spans subdivisions 15A–17B, and it is present in the standard arrangement in the AatrE4 assembly according to the *An. atroparvus* cytogenetic map (Fig. 3f).

Our Hi-C maps revealed breakpoints for two polymorphic inversions in species of the *An. gambiae* complex, *An. coluzzii* and *An. merus* (Supplementary Fig. 2C, D). The cytogenetic position of the inversion identified in *An. coluzzii* corresponds to the known polymorphic inversion 2Ru[27,37]. The MOPTI strain of *An. coluzzii*, from which we obtained the Hi-C data, is known to be polymorphic for the 2Ru inversion (see description at https://www.beiresources.org/Catalog/BEIVectors/MRA-763.aspx). However, the genomic coordinates of this inversion have not been identified. Here, we report the approximate genomic coordinates of the 2Ru inversion breakpoints in our assembly (Supplementary Table 3 and Supplementary Fig. 2C). Our dot plot alignment of the AcolN2 genome assembly with the AgamP4 assembly and

with the recently sequenced genome AcolMOP1 of the *An. coluzzii* MOPTI strain[38] demonstrated that all these assemblies have the standard arrangement with respect to the 2Ru inversion (Supplementary Fig. 1A, B). However, unlike AgamP4 but similarly to AcolMOP1, AcolN2 has the inverted 2La arrangement (Supplementary Fig. 1A). In addition, we discovered a novel small 2.8 Mb polymorphic inversion near the pericentromeric region of the 2R arm in *An. merus*. We named the inversion 2Rt and mapped its breakpoints with a 5 kb precision with the current depth of sequencing (Supplementary Table 3 and Supplementary Fig. 2D). The dot plot alignment with AcolN2 showed that the AmerM5 assembly captured the standard arrangement with respect to this inversion (Supplementary Fig. 1C). Thus, Hi-C is a robust tool for visualization and discovery of chromosomal inversions in malaria mosquitoes.

**3D-chromatin structure of *Anopheles* genome revealed by Hi-C.** The Hi-C contact maps clearly delineated five chromosomal arms (elements) for all experimental genomes—X, 2R, 2L, 3R, 3L (Fig. 4, A shows an example of chromosomal elements of *An. albimanus*). On average, interactions within chromosomal elements (cis-interactions) were 4.9 times more frequent than interactions between chromosomal elements (trans-interactions). Notably, we found that the chromosome X had a decreased cis-to-trans interaction ratio compared to autosomes (Supplementary Fig. 3).

Another upshift of trans interactions was observed in centromeric and telomeric regions of all chromosomal elements

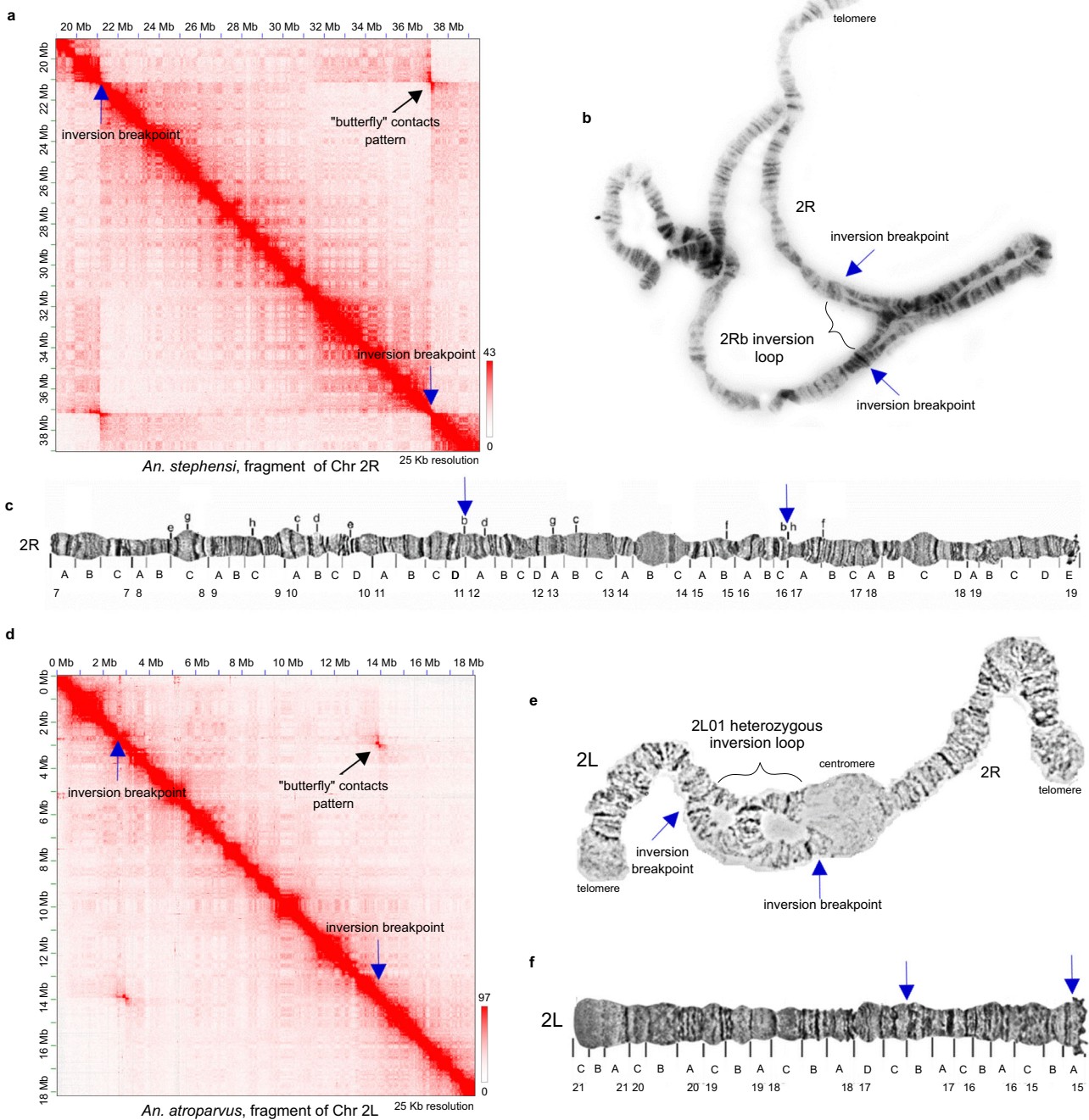

**Fig. 3 Polymorphic inversions identified using Hi-C data. a** Region of the *An. stephensi* Hi-C contact map with the 2Rb polymorphic inversion manifested as a clear "butterfly" pattern, color bar reflects the contact frequency; **b** Light microscope image of *An. stephensi* polytene chromosomes showing the 2Rb inversion heterozygous loop; **c** The cytogenetic map of *An. stephensi* chromosome 2R, published previously in Jiang et al.[25]; **d** Region of the *An. atroparvus* Hi-C contact map with the 2L1 polymorphic inversion, color bar reflects the contact frequency; **e** Light microscope image of *An. atroparvus* polytene chromosome 2 showing the 2L01 heterozygous inversion loop; **f** Physical map of *An. atroparvus* chromosome 2L, published previously in Artemov et al.[23]. Inversion breakpoints are shown with blue arrows in all panels.

(Supplementary Fig. 3). Strong centromere-centromere clustering as well as inter-chromosomal and intra-chromosomal telomere–telomere interactions were detected. These contact patterns suggest a Rabl-like configuration[39–41] in *Anopheles* genomes where centromeres and telomeres form clusters within the nuclear space (Fig. 4b). Additionally, we observed another manifestation of the Rabl-like configuration represented by interactions between chromosomal arms, which is seen as "wings" located perpendicular to the main diagonal on the Hi-C contact map (Fig. 4a). Quantitative analysis of contact frequencies confirmed an increase of interactions between loci from different chromosome arms located equidistantly from the centromeres (Supplementary Fig. 4). This increase of interactions was more pronounced in embryonic tissues (1.49–2.45 times) than in adult mosquito data (1.19–1.25 times), suggesting that the Rabl-like configuration might be a feature of some but not all cell types. 3D-FISH experiments on *An. stephensi* ovarian tissue confirmed a tight clustering of the centromeric regions in follicle cells and the lack of such tight clustering in nurse cells within polytene chromosomes (Fig. 4c).

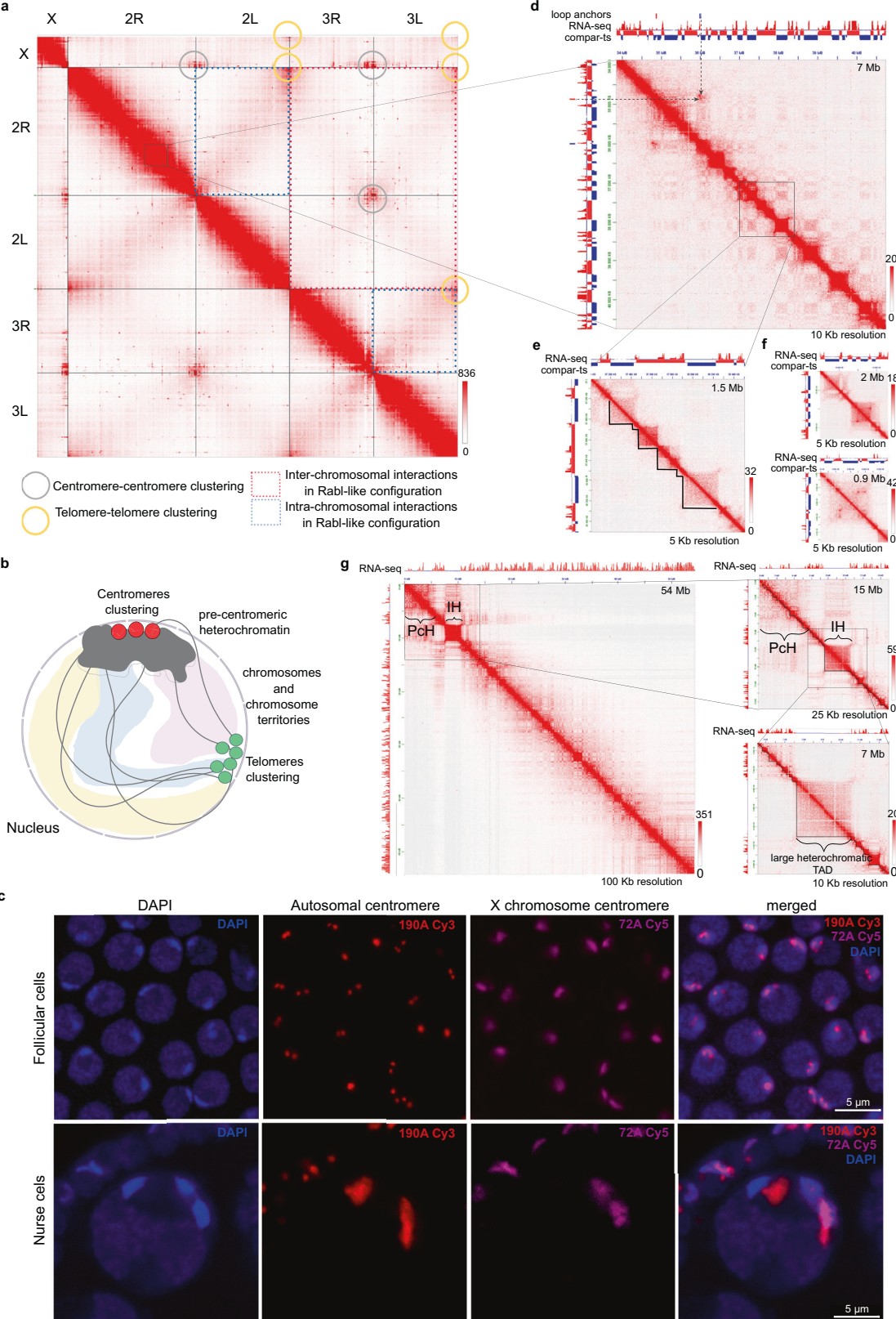

**Fig. 4 3D chromatin structure of *Anopheles* mosquito genomes. a** A Hi-C contact map for *An. albimanus*; gray and yellow circles correspond to contacts between telomeres and centromeres, respectively; **b** Schematic representation of Rabl-like configuration; **c** FISH showing position of centromeres: 190A Cy3 probe—autosomal centromere, 72A Cy5 probe—Chr X centromere; *n* = 144 for follicular cells, *n* = 8 for nurse cells; **d**–**g** Zoomed-in Hi-C map regions showing long-distance loops between TADs (**d** indicated by arrows), compartments (**d**), TADs (**e**), loops within TADs (**f**), and heterochromatic blocks either located near centromeres (PcH) or interspersed throughout the euchromatin (IH) (**g**). Color bars near the heatmaps reflect the contact frequency.

Using zoom-in views of the contact maps, we observed a checkerboard-like pattern of long-range interactions, which was previously shown to represent spatial compartments of the chromatin (Fig. 4d). Some long-range contacts were extremely pronounced, corresponding to looping interactions between loci located several Mbs away from each other (Fig. 4d and Supplementary Data 3). At the highest resolution (1–25 kb) we observed triangles above the diagonal, corresponding to the TADs (Fig. 4e), and chromatin loops located within some of these TADs (Fig. 4f). Magnified views of the contact maps also revealed large insulated blocks of chromatin located either near centromeres or interspersed throughout chromosomal arms (Fig. 4g and Supplementary Fig. 5A–F). The sizes and locations of these blocks are in agreement with the positions of pericentromeric and intercalary heterochromatic blocks on standard cytogenetic maps[42] (Supplementary Fig. 6). We next aimed to quantitatively characterize compartments, TADs, and loops identified in *Anopheles* genomes.

**Anopheles genomes are partitioned into compartments.** Inspecting genomic interactions at various resolutions, we observed prominent signs of genome compartmentalization, manifested as plaid-patterns of chromatin contacts on the Hi-C maps (Fig. 4). However, when we employed a principal component analysis (PCA), which is widely used to identify spatial chromatin compartments[1,43], we found that the first principal component (PC1) of the Hi-C matrix does not reflect the observed plaid-patterns for the majority of chromosomes (Fig. 5a, b and Supplementary Fig. 7A–I). In most cases, chromosomal arms were split into two or three large, contiguous regions characterized by similar PC1 values. Thus, PC1 values reflect the position of the locus along the centromere–telomere axis rather than the plaid-pattern of contacts.

This discordance between compartmental interactions and PC1-values could be explained by Rabl-like configuration of chromosomes, i.e., clustering of large blocks of centromeric and telomeric heterochromatin and/or elongated shapes of chromosome territories. First, clustering of centromeric and/or telomeric chromatin may dominate the clustering of compartments. This explains those cases where centromeric and telomeric regions display similar PC1 values, whereas the rest of the chromosomal arms display opposite PC1 values (Supplementary Fig. 7A–I). Second, it has been shown in *Drosophila* cells that a Rabl-like configuration results in elongated shapes of the chromosome territories[44], preventing clustering of actively expressed genes located far from each other on the centromere-telomere axis.

Analysis of PC1 and PC2 distributions (Fig. 5c) showed a pattern very similar to those observed previously in barley[45], a plant species with Rabl-like configuration of chromosomes. Similar to mosquitoes, PC1 and PC2 values in plants reflect the position of the locus on the centromere-telomere axis. This additionally supports the link between Rabl-like chromosome configuration and specific distribution of PC1/PC2 values.

Moreover, for adult *An. merus* samples, where signs of a Rabl-like configuration of chromosomes were much less pronounced (Supplementary Fig. 4), the standard PC1 algorithm was able to define compartments that agree with plaid-patterns on all chromosomes (Supplementary Fig. 7K–M). Thus, we suggest that the Rabl-like configuration of chromosomes attenuates long-range interactions between compartments.

To define compartments in mosquito genomes, we attempted either to exclude centromeric and telomeric regions from PCA (cropping approach) or computing local PC1 in megabase-scaled frames (framing approach). However, neither cropped nor framed data does not allow capturing of the observed plaid-

pattern of contacts (Supplementary Note I). Therefore, we developed a novel approach for robust identification of compartments on chromosomes with Rabl-like configurations. Briefly, our approach relies on 1) local smoothing which reduces noise in long-range interactions; 2) normalization of Pearson's correlation values within small blocks of the data matrix; and 3) PC1 calculation (see Supplementary Note I for details and comparisons of different approaches). We called this algorithm "contrast enhancement" because the result of Pearson's correlation scaling within small blocks resembles enhancement of image contrast (Supplementary Fig. 8). We refer to the obtained PC1 values as cePC1 (contrast-enhanced PC1 values). Using contrast enhancement we were able to identify compartments that correspond well to the plaid-pattern observed on the Hi-C maps (Fig. 5b).

To understand epigenetic mechanisms underlying *Anopheles* chromatin compartments, we performed RNA-seq on the same embryonic stages as used for Hi-C library preparation. We found that compartments show moderate correlation with expression levels (Fig. 5d), slightly lower correlation with gene density, and only weak correlation with GC-content (Supplementary Data 4). Correlations were substantially higher for cePC1, obtained by contrast enhancement, whereas for the original PC1 values had almost no correlation with the aforementioned epigenetic features (Supplementary Data 4). Based on the cePC1 values, we split the genome into two non-overlapping A-compartment and B-compartment so that loci belonging to the A-compartment showed higher gene density, gene expression, and GC-content.

Overall, we have shown that *Anopheles* genomes are partitioned into distinct chromatin compartments. The compartmentalization is weakened in embryonic cells presumably due to Rabl-like configuration of chromosomes. We developed a computational approach to detect compartments and showed that spatial compartmentalization distinguishes the active A-compartment (gene dense, actively expressed) and inactive B-compartment (gene-poor, mostly silent).

**Transitions between compartments correspond to TAD boundaries in Anopheles genomes.** In addition to the plaid pattern, Hi-C maps display a symmetric triangle pattern on each side of the main diagonal. These triangles represent chromatin interactions within TADs (Figs. 4 and 6a, b). For each species, we quantified genome-wide insulation levels and we delineated TADs at 5 kb resolution using insulation score values. The resulting TADs range in size from 15 to 650 kb, with a median size of ~135 kb (Supplementary Table 4). Distribution of TAD sizes was similar for all chromosomal elements (Figs. 6c and 9a). However, within each chromosome arm, smaller TADs tend to co-localize with A-compartment (euchromatic) regions, whereas longer TADs co-localize with B-compartment (heterochromatic) regions (Fig. 6d and Supplementary Fig. 9B). As evident from the analysis of cePC1-values, TADs longer than 400 kb are almost exclusively located in gene-poor, heterochromatic regions with a low level of gene expression. Comparing positions of the largest TADs with cytogenetic maps of *Anopheles* polytene chromosomes[42], we found that these chromatin structures often correspond to the regions of intercalary heterochromatin (Fig. 6e and Supplementary Fig. 6) and demonstrate a reduction in gene density and in RNA-seq signal. A previous work identified two types of intercalary heterochromatin regions in *An. gambiae*— diffuse and compact heterochromatin[42]. Here, we show that diffuse intercalary heterochromatin forms B-compartments with regions of pericentromeric heterochromatin but not with euchromatic regions. Compact intercalary heterochromatin forms weak B-compartments with euchromatic regions, but not with other heterochromatic regions. Overall, heterochromatic TADs

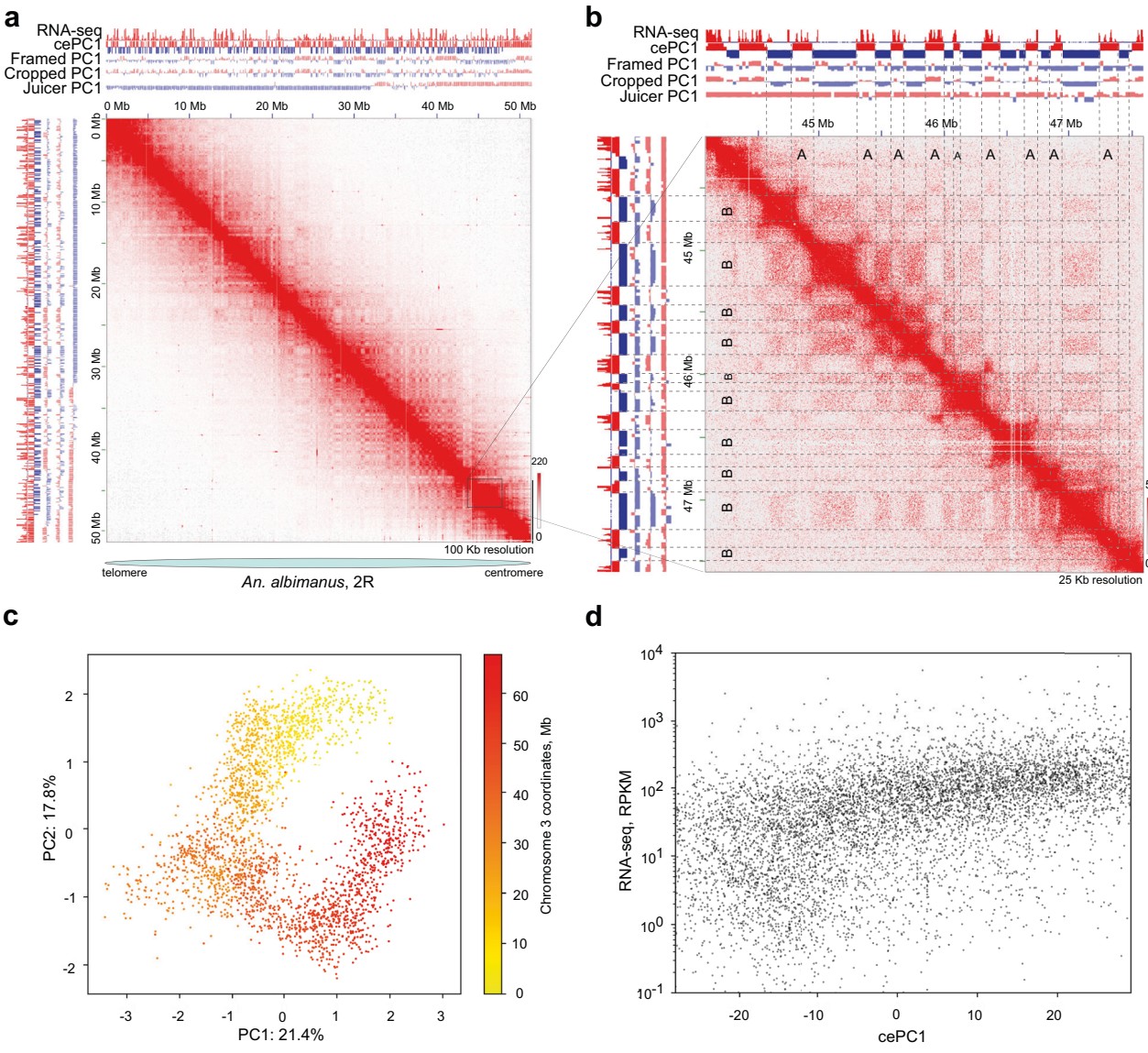

**Fig. 5 *Anopheles* genomes are partitioned into compartments. a** Compartmentalization demonstrated for Chr 2R, *An. albimanus*; **b** Region zoomed-in from Fig. 5, **a** correspondence between RNA-seq data and PC1 values, obtained using different algorithms; color bars on **a**, **b** panels reflect the contact frequency; **c** Scatter plot of PC1 vs. PC2 values with color representing the coordinates of loci on Chr 3 of *An. albimanus* illustrates the dependence of PC values from genomic location. Each point represents a 25 kb bin; **d** cePC1 correlation with RNA-seq data ($R = 0.67$). Each point represents a 25 kb bin. Negative cePC1 values represent B-compartments.

do not form strong long-range interactions, typical of smaller TADs (Supplementary Fig. 9E, F).

We next analyzed TAD boundaries and found that in the majority of cases the boundaries coincide with the transitions between the compartments (Fig. 6f and Supplementary Fig. 9C), and the magnitude of cePC1 changes was higher around TAD boundaries than within TADs ($p$-value < 10E−30 for all species, Supplementary Fig. 9D). Consistent with this, we found that insulation score correlates with the density of promoters (identified as gene starts), cePC1 values and transcription (Supplementary Table 5). We additionally inspected all cases where a strong TAD boundary (top quartile of insulation score distribution) separated regions with similar cePC1 values (cePC1 difference belongs to the bottom quartile of the distribution). Some of these cases appeared to be due to different resolutions of cePC1 vector (computed at 25 kb binned matrix) and TAD-boundaries (called at 5 kb resolutions). We also found a limited number of wrong cePC1 calls.

Thus, we concluded that both TADs and compartments represent similar chromatin features; local insulation from the neighboring genomic regions is captured by TAD-callers, whereas preferences between long-range interactions of these locally insulated genomic regions are captured by cePC1. Chromosomal regions of intercalary heterochromatin correspond to the large TADs depleted of long-range interactions.

**Hi-C maps identified long-range interactions and FISH validated chromatin loops.** Examination of the Hi-C data revealed looping interactions between specific loci (Fig. 4d, f and Supplementary Fig. 10A–H). Many of these interactions occurred within the same TADs, and genomic regions involved in looping (hereinafter referred as loop anchors) often formed networks of interactions (Supplementary Fig. 10A, B). As chromatin loops were previously associated with the presence of polycomb repressive complexes[13], we performed H3K27me3 ChIP-seq on *An. atroparvus* and found that indeed anchors of some loops are

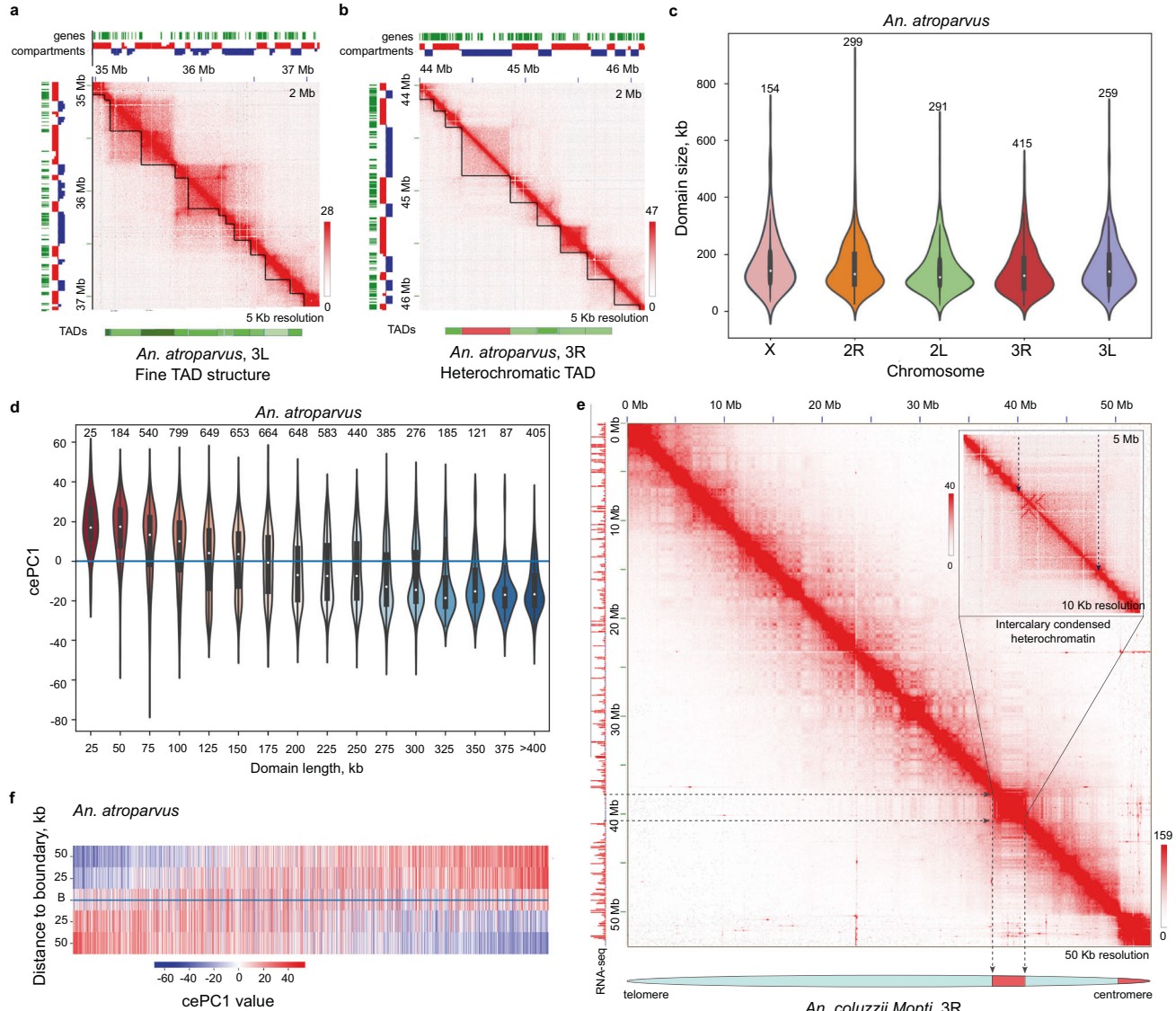

**Fig. 6 TADs in *Anopheles* genomes. a** Typical TADs structure; **b** Example of heterochromatic TAD (marked in red below the contact map region); **c** TADs size distributions within the chromosomes, total number of TADs: $n = 1418$; **d** Violin-plot illustrating the distributions of cePC1-values stratified by TADs length. Center and bounds of box correspond to Q1, median, and Q3; whiskers extend between Q1–1.5IQR and Q3 + 1.5IQR; **e** Compact intercalary heterochromatic block visualized on the Hi-C map of Chr 3R *An. coluzzii*; **f** Heatmap of cePC1 values around TAD boundaries. Each line represents one TAD boundary. **a–d**, **f** show data for *An. atroparvus*. Data for other species are shown in Supplementary Fig. 9. Color bars on **a**, **b**, **e** panels reflect the contact frequency.

located within H3K27me3-enriched regions (Supplementary Fig. 10A–E) but not others (Supplementary Fig. 10F–H).

Whereas the majority of loops spanned genomic distances of less than 1 Mb and occurred within TADs, we found notable examples of extremely long loops connecting loci separated by up to 31 Mb (Fig. 7a–e and Supplementary Fig. 11A–E). We found from 2 to 6 such giant loops in each species (Supplementary Table 3) and decided to investigate two of them in greater details (Supplementary Table 6). One such loop is located on chromosome X (X-loop) (Fig. 7a–e) and the other loop is located on an autosome (A-loop) (Supplementary Fig. 11A–E).

Anchors of these loops were represented by large (~200–300 kb) loci, interacting significantly more than expected at the specific genomic distance (top 1–2% of all interactions) (Supplementary Fig. 12A, C, E, G, I and Supplementary Table 6). Within long anchors of the loops, we often observed two pairs of smaller loci (~25 kb) showing high interaction frequencies (Fig. 7f and

Supplementary Fig. 11F). We noticed that the X-loop anchors do not interact with A-loop anchors above the level of average interchromosomal contacts, with only one exception found in *An. albimanus*, where interaction between the right anchor of the X-loop and the left anchor of the A-loop was among the strongest 6% of interchromosomal interactions. However, other pairs of anchors of these loops (left/left, left/right, right/right) did not show strong enrichment (0th, 54th, and 64th percentiles) suggesting that the X-loop and A-loop anchors do not colocalize in the nuclei space (Supplementary Fig. 12 and Supplementary Table 6).

To understand whether loops identified in different *Anopheles* species are homologous to each other, we compared the genes located within the loop anchors. Surprisingly, four of five species (all except *An. albimanus*) shared orthologous genes within the X-loop anchors (Supplementary Data 5) and all the species had orthologous genes within the A-loop anchors.

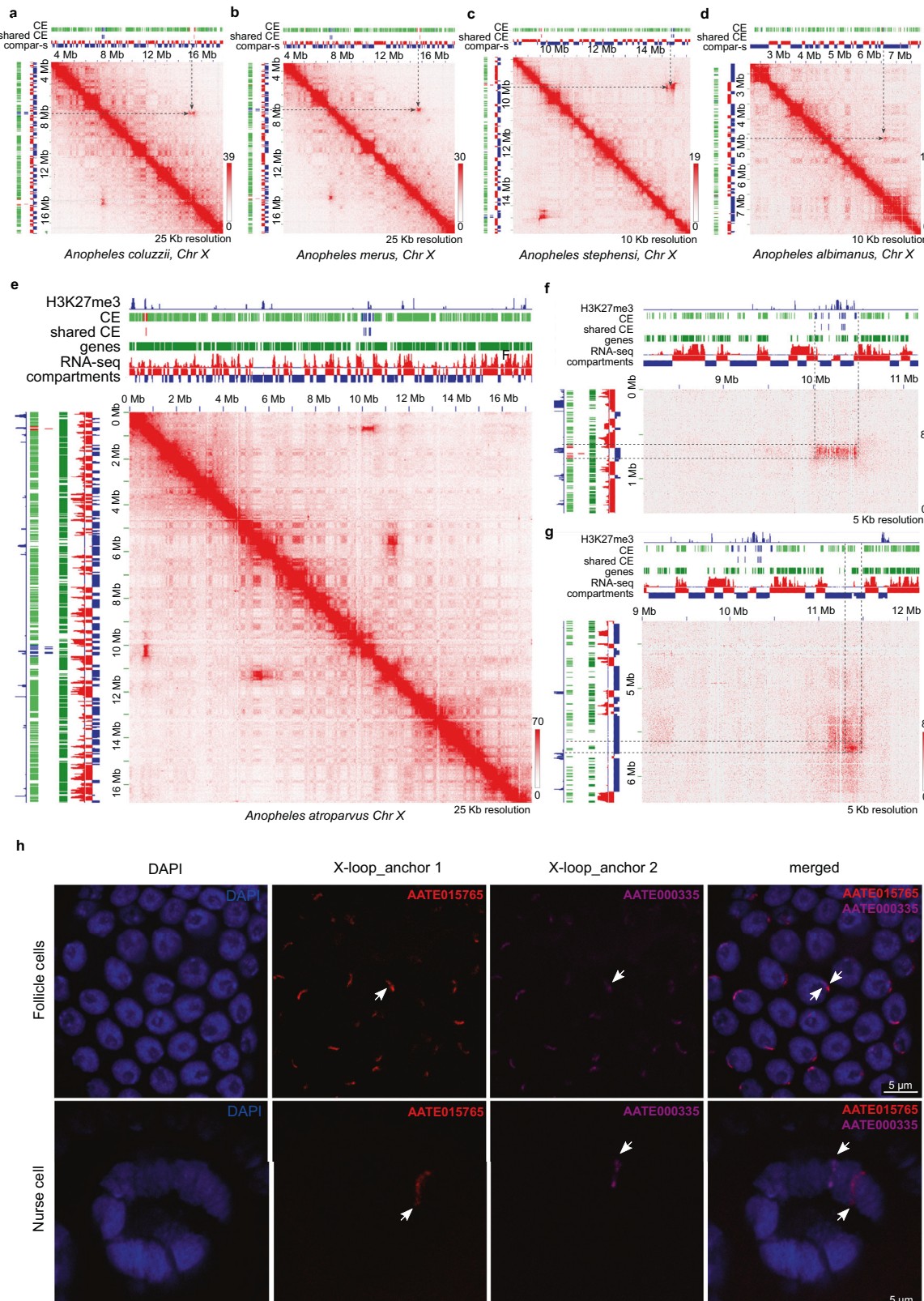

**Fig. 7 Long-range interactions identified on Hi-C maps and loop validation by FISH. a–d** X-loop in *An. coluzzii* (**a**), *An. merus* (**b**), *An. stephensi* (**c**), *An. albimanus* (**d**); **e** Region of *An. atroparvus* chromosome X Hi-C map, showing the X-loop and another giant loop identified on the X chromosome; **f**, **g** Zoomed-in regions of two loops identified on the X chromosome in *An. atroparvus*. Color bars near the heatmaps reflect the contact frequency. The red tracks are CEs within the first anchor of the loop. The blue tracks are CEs within the second anchor of the loops. The green tracks are the other CEs. **h** Colocalization of X-loop (showed on **g**) anchors identified by 3D FISH in follicle cells but not in nurse cells of *An. atroparvus*.

Next, we performed whole-genome alignments to identify conserved elements (CEs) and compared CE distribution within loop anchors. We used CEs located within the *An. atroparvus* loop anchors to determine the anchor synteny between species. The results confirmed that loops are formed between homologous (syntenic) regions (Fig. 7a–c, e and Supplementary Fig. 11A–E), except for the left anchor of the X-loop in *An. albimanus* (Fig. 7d), which does not share a CE with X-loop anchors in other species. Several of the CEs were shared between all the species. However, at least in some species the shared CEs were located outside of the subregions of anchors displaying the highest interaction frequencies suggesting that none of the identified CEs could explain loop formation.

We also analyzed gene expression and H3K27me3 profiles within loop anchors. Several orthologous genes were located within anchors of all examined species (Supplementary Data 5). Some of them were expressed, although not all anchors contained actively expressed genes (Fig. 7f, g and Supplementary Fig. 11F) and the expression level was moderate. In accordance with this observation, the cePC1 analysis showed that loop anchors were enriched in B-compartment (Fig. 7a–g and Supplementary Fig. 11A–G). Thus, long-range X-loops and A-loops could not be explained by the clustering of active chromatin.

The H3K27me3 signal was previously associated with Polycomb-mediated loops[13]. We compared the H3K27me3 signal in the *An. atroparvus* X-loop and A-loop anchors with a background genomic signal and with the signal observed in a manually curated set of intra-TAD loops located within H3K27me3-rich loci (Supplementary Data 6). This analysis showed a moderate H3K27me3 enrichment within the X-loop anchors and no enrichment for the A-loop anchors (Supplementary Fig. 13), which contrasted with the high enrichment of the H3K27me3 signal within the intra-TAD loop anchors. Additionally, we observed several chromatin blocks highly enriched in H3K27me3 histone mark and located between X-loop anchors that do not form any long-distance loops. We concluded that Polycomb-mediated interactions might be implicated in the formation of the X-loop, but not the A-loop. Moreover, the H3K27me3 enrichment could not explain X-loop formation solely, because other regions with similar H3K27me3 enrichment levels did not show the same increase of spatial interactions.

We compared Hi-C maps obtained from adult and embryonic *An. merus* tissues and found that A-loops and X-loops are present at both developmental stages (Supplementary Fig. 14A, B). This indicates that loops are not specific to one particular developmental-stage or cell type.

Finally, we performed 2D and 3D FISH to validate the interactions between the putative loop anchors in nuclei of follicle cells and ovarian nurse cells of *An. coluzzii*, *An. stephensi*, and *An. atroparvus*. In fact, in highly polytenized chromosomes of nurse cells we found no colocalization of probes at putative anchors in all tested species by both 2D and 3D FISH. For the *An. coluzzii* 7.5 Mb X-loop, we found 100% colocalization of probes at anchors in follicle cells by 2D FISH. For the *An. stephensi* 5.5 Mb A-loop, almost no colocalization was identified in follicle cells by 3D FISH (8%, $n = 74$), but the anchors colocalized with high frequency (61% of nuclei, $n = 157$) in ovarian nurse cells with low-level polytene chromosomes (Supplementary Fig. 11G). We additionally tested three loops of different sizes in *An. atroparvus*: 31 Mb A-loop on arm 3R, 12 Mb loop on arm 2R, and 6 Mb loop on chromosome X (not the conserved one). The anchors were always colocalized in follicle cells in 2D FISH experiments. However, colocalization in 3D FISH experiments was variable: from partial colocalization for the large 31 Mb A-loop on arm 3R (12%, $n = 109$) and medium 12 Mb loop on arm 2R (15%, $n = 55$) to almost constitutive colocalization for the smaller 6 Mb

loop on chromosome X (97%, $n = 60$). Colocalization for the 31 Mb A-loop was also confirmed by 3D FISH in low-level polytenized chromosomes (30%, $n = 10$) but not in highly polytenized chromosomes of nurse cells of *An. atroparvus* by 3D FISH (Fig. 7h). Genomic coordinates for all loop anchors mentioned above and complete FISH statistics can be found in Supplementary Data 3 and Supplementary Data 7, respectively.

Overall, we validated most of the long-range interactions detected by Hi-C using FISH, although colocalization of loop anchors was found only in a subset of examined cell types. The results show that *Anopheles* chromatin forms several extremely long-range, locus-specific contacts, which are evolutionarily conserved for ~100 million years and based on currently unknown molecular mechanisms independent of active transcription or Polycomb-group proteins.

**Evolutionary comparison of spatial genome organization in five *Anopheles* species.** Profiling chromatin interactions in several species across the *Anopheles* genus allowed us to study how nuclear architecture contributes to genome evolution. We aimed to understand 1) whether evolutionary breakpoint regions (EBRs), i.e., boundaries of homologous synteny blocks, preferentially occur near TAD boundaries and 2) whether TAD and compartment boundaries are conserved within synteny blocks. To answer these questions we defined EBRs and synteny blocks, and determined lineage-specificity of EBRs based on phylogenetic relations between species (Fig. 8a). Note that for this analysis we excluded centromeric and telomeric heterochromatin, because we can not guarantee the accuracy of EBR calls and other genomic features in these regions. We further defined *reused* breakpoints (marked with circumflex "^") as EBRs observed in two lineages but not their common ancestor with respect to an outgroup, and considered all other EBRs as *lineage-specific* (marked with asterisk "*", see Fig. 8a for details). So that, hereinafter each class of EBRs represents a different lineage of breakpoint origin.

To access common properties of EBRs, we characterized their genomic landscapes. We found that synteny breakpoints occur more often in the euchromatic regions belonging to the A-compartment than in the B-compartment (Fig. 8b and Supplementary Fig. 15). Whereas distribution of cePC1 values at random points of the genome was bimodal, almost all classes of EBRs showed unimodal distribution of the cePC1 values with the median shifted towards positive cePC1 scores. This enrichment of the A-compartment in EBRs was especially pronounced for reused breakpoints.

Consistent with enrichment of the A-compartment in EBRs, we found strong enrichment of gene density around EBRs (Fig. 8c). Despite the increased gene density around EBRs, there was a strong depletion of EBRs located within gene bodies suggesting negative selection acting against disrupting gene sequences (Supplementary Fig. 16).

Repetitive elements have been implicated in generating chromosomal rearrangements in fruit flies and mosquitoes[32,46]. To assess a landscape of transposable elements and simple repeats around EBRs, we performed de novo repeat identification and classification. We found that repetitive elements comprise 4–20% of the assembled genomes and the X chromosome displayed a higher repeat content than autosomes (Supplementary Data 8). Analysis of repetitive element distributions showed that all studied repeat classes were uniformly distributed and not enriched around TAD boundaries or EBRs (Supplementary Figs. 17–19).

We next aimed to compare the location of EBRs and TAD boundaries. Using an insulation score as a proxy for TAD

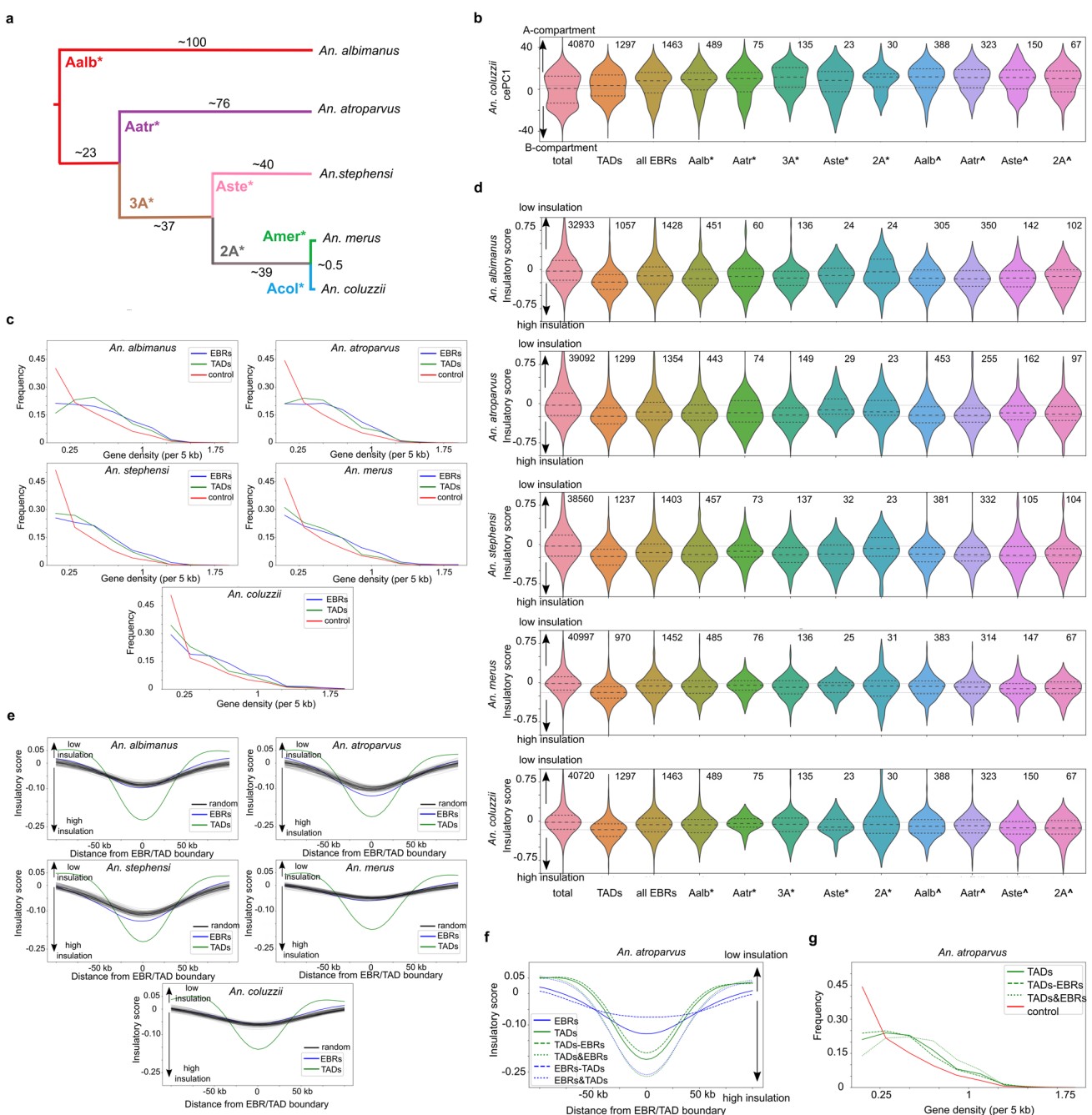

**Fig. 8 Evolutionary comparison of spatial genome organization in five *Anopheles* species. a** Phylogeny showing EBR classes; 3A* corresponds to rearrangements that occured in the common ancestor of three *Anopheles* species (*An. coluzzii*, *An. merus*, and *An. stephensi*); 2A* corresponds to rearrangements shared by *An. coluzzii* and *An. merus*; **b** cePC1 value distributions in the *An. coluzzii* genome. Labels on the X axis correspond to the different EBR classes where reused breakpoints are marked with circumflex "^" and lineage-specific breakpoints with asterisk "*". Violin-plots show genome-wide distribution (total), distribution observed at TAD boundaries (TADs) followed by distributions for each class of EBRs. Numbers of EBRs are shown above each violin-plot; **c** Gene density at EBRs; *Y*-axis shows percentage of genomic bins for each gene of the density group depicted on *X*-axis; **d** Insulation of TAD boundaries and EBRs. Data are shown as in **b**; **e** Insulation of EBRs (blue line) and TAD boundaries (green line) compared to randomized controls with gene density matching EBRs (black lines); **f**, **g** Insulation (**f**) and gene density (**g**) for regions containing both TAD boundaries and EBRs (TADs&EBRs) and containing only TAD boundaries (TADs-EBRs) or only EBRs (EBRs-TADs). *Y*-axes show insulation (**f**) or percentage of genomic bins for each gene density group (**g**).

boundaries, we showed that all classes of EBRs preferentially occurred in insulated regions (Fig. 8d). To better understand this EBR feature, we compared the insulation landscape of different EBR classes.

We note that most chromosomal rearrangements were fixed during evolution in ancestral species, whereas we measured

insulation and other genomic properties using genomic data of extant species. Thus, we expected that if EBR positions depend on genome topology, this dependence should be more pronounced for those chromosomal rearrangements in the current lineage that occurred more recently. This assumption was indeed valid for some cases. For example, when measuring insulation in *An.*

*albimanus*, we found that positions of breakpoints that occurred in the *An. albimanus* and *An. atroparvus* lineages (*Aalb\** and *Aatr\** in Fig. 8d) were characterized by stronger insulation than positions of breakpoints that occured in *An. stephensi* (*Aste\**) or shared between *An. merus* and *An. coluzzii* (*2A\**) lineages. At the same time, positions of *An. stephensi* specific breakpoints (*Aste\**) and breakpoints shared between *An. coluzzii, An. merus* and *An. stephensi* (*3A\**) showed very high insulation in *An. stephensi*, reaching the level of the TAD boundary insulation. Regions of reused breakpoints were in many cases highly insulated reaching the level of the TAD boundary insulation in some datasets, for example, when *An. stephensi* was used as a reference.

However, there were many examples when the insulation of EBRs did not correlate with the phylogenetic relationships between lineages. This can be demonstrated, for example, by using the *An. stephensi* genome as a reference and comparing insulation at *Aalb\**, *Aatr\**, and *Aste\** EBRs. Relative to the *An. stephensi* reference, *Aalb\** EBRs occurred ~100 million years ago in a different lineage, *Aatr\** EBRs occurred 40–80 million years ago in different lineages, and *Aste\** EBRs occurred in the reference lineage no more than 40 million years ago. Thus, one would expect that *An. stephensi* genomic loci orthologues to *Aste\** EBRs are the closest to ancestral states and display the highest insulation, followed by insulation of loci orthologues to *Aatr\** EBRs, whereas *Aalb\** EBRs should display the lowest insulation. However, this was not the case, because two classes of EBRs (*Aalb\** and *Aste\**) displayed relatively similar insulation levels, whereas *Aatr\** EBRs showed decreased insulation. Similarly, *Aalb\** EBRs showed about the same or even slightly higher insulation than *An. atroparvus* (*Aatr\**) or *An. stephensi* (*Aste\**) EBRs when using the *An. merus* genome as a reference, even though the latter two lineages are closer to the reference.

In general, we found that differences in the levels of insulation between EBR classes are inconsistent with the phylogenetic relationship between lineages. The only pattern which holds in almost all cases is that insulation of EBRs (irrespective of their lineage of origin) was stronger than genome average level, suggesting non-random positioning of EBRs with respect to the genome topology.

We also noted that some features of EBRs and TAD boundaries are similar. For example, both EBRs and TAD boundaries occurred in the A-compartment more often than in the B-compartment (Fig. 8b). Gene density was also very similar for EBRs and TAD boundaries (Fig. 8c) significantly exceeding genome-wide average level.

Based on these observations, we speculated that some of these genomic features may represent a confounder that guides the genomic location of both TAD boundaries and EBRs. To test this hypothesis, we generated randomized control regions so that the distribution of specific genomic features matched the distribution observed for EBRs. For each of the obtained randomized samples, we computed insulation scores and compared them with the insulation scores observed for EBRs.

When using random regions with the distribution of A/B-compartments or repetitive elements (either simple repeats, DNA-transposons, or retrotransposons) matching the distribution of these features in EBRs, we did not see any increase of insulation compared to the genome-wide average (Supplementary Figs. 20–23). However, when we generated random regions with gene density distributions matching those observed for EBRs, we found that these random regions showed increased insulation (Fig. 8e).

Notably, the insulation of random regions sampled this way did not differ from the insulation level observed in EBRs for *An. albimanus, An. merus*, and *An. coluzzii*. For *An. atroparvus* and *An. stephensi* EBRs were more insulated than control regions, but

this difference between EBRs and random regions was small. Thus, the occurrence of EBRs in gene-dense regions could explain the overlap between TAD boundaries and synteny breakpoints, requiring no additional constraints on EBR location caused by genome architecture itself.

We next asked whether those TAD boundaries that co-occur with EBRs have any specific properties. We compared gene density and insulation between three groups of regions: TAD boundaries overlapping EBRs (TAD&EBRs in Fig. 8f, g), TAD boundaries non-overlapping EBRs (TAD-EBRs), and EBRs non-overlapping TAD boundaries (EBR-TADs). As expected, we found that EBRs overlapping TAD boundaries gain insulation compared to EBRs not overlapping TAD boundaries (Fig. 8f and Supplementary Fig. 24). Moreover, TADs overlapping EBRs displayed higher insulation as well as slightly higher gene density compared to other TAD boundaries (Fig. 8g and Supplementary Fig. 25). These observations are in accord with our hypothesis that gene-dense regions are enriched for both highly insulated TAD boundaries and evolutionary-fixed chromosomal rearrangements.

Because a large portion of EBRs occurred within TADs, we asked whether there are any specific properties of TADs which are reshuffled in the course of evolution. We compared the insulation of TADs containing at least one EBR and TADs containing no EBRs. We found that TADs containing EBRs show slightly higher insulation; however, we also noted that longer TADs are on average more insulated than short TADs. Thus, the difference in length explains higher EBR frequency of highly insulated TADs. Overall, we concluded that EBR occurrence is largely independent from genome architecture in *Anopheles* mosquitoes.

It might be that EBRs occur randomly, however the regions joined together as a result of EBRs share certain common properties. To test this hypothesis, we defined the ancestral state for each EBR as the chromatin state of the closest species with the ancestral order of genomic segments (Fig. 9a). Phylogenetic analysis allowed us to find ancestral states for approximately 3200 EBRs for *An. albimanus* and 400–650 EBRs for the other *Anopheles* species. We then focused on the ancestral cePC1 values (reflecting A/B-compartments) of regions, which are joined together by evolutionary fixed chromosomal rearrangements. The probability of switching the compartment of the genomic neighbor does not differ from the probability in the control dataset obtained by random joining ends of synteny blocks (Fig. 9b and Supplementary Fig. 26). Similarly, we found no dependence between the compartmental states of two regions joined by chromosomal rearrangements (Fig. 9b; Supplementary Fig. 26A, B). To sum up, our data did not demonstrate any preference to preserve epigenetic states of neighboring loci in the course of chromosomal evolution.

We next aimed to understand how genome architecture evolves within synteny breakpoints. We performed interspecies liftover of Hi-C maps using C-InterSecture tool and computed similarity scores for each pair of species[11]. We found that chromatin contacts differ between species significantly less than expected at random, indicating evolutionary conservation of genome organization within synteny blocks (Fig. 9c). However, the level of conservation was relatively similar for all pairwise species comparisons (except comparison between *An. coluzzii* and *An. merus*, which showed a very high level of similarity), and did not reflect evolutionary distances between species (Fig. 9c).

In accordance with genome-wide statistical analysis, visual inspection of liftovered Hi-C maps showed high similarity of contact patterns (Fig. 9d, e). For example, systematic inspection of *An. coluzzii* and *An. stephensi* maps resulted in only a single example of a syntenic region with clear difference in contact pattern between these species (Fig. 9e). Even for very distant species, such as *An. coluzzii* and *An. albimanus*, visual inspection

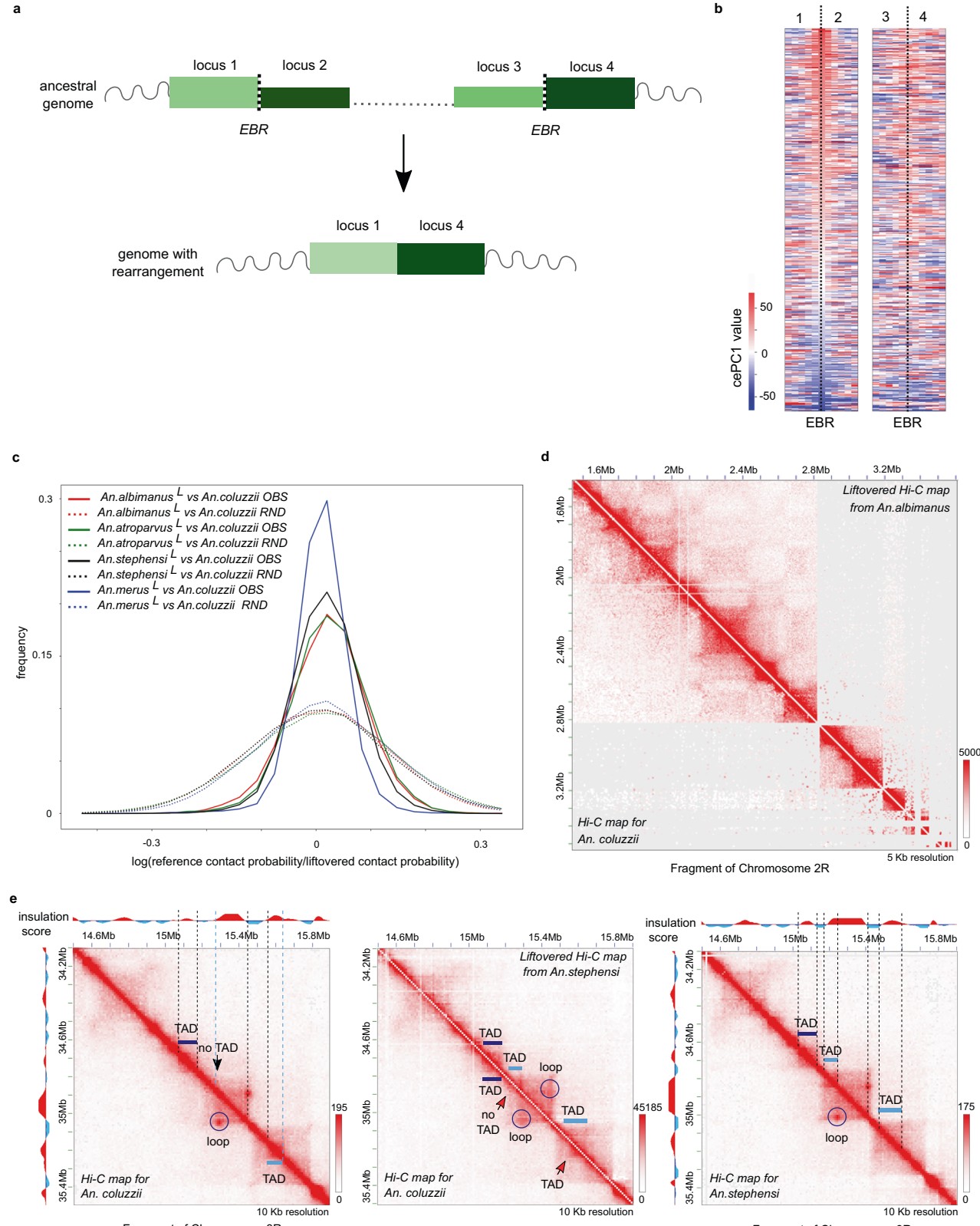

of few long syntenic blocks showed remarkable level of similarity of genomic contacts (Fig. 9d).

**Spatial contacts of the chromatin in insects and vertebrates are constrained in a genome-size-dependent manner.** To study general principles of genome organization in *Anopheles* we analyzed how chromatin contact probability (P) scales with genomic

separation s, P(s). As reported in previous studies, contact frequencies decay rapidly as genomic distance increases (Fig. 10a). Since it was shown previously that P(s) follows a power law, we characterized the exponent by computing the slope of the decay curve in log–log coordinates (Fig. 10b). We found that decay speed is not uniform, and could be described by two different decay phases.

**Fig. 9 3D-genome organization conserved within syntenic blocks. a** Scheme showing reordering of genomic segments as a result of evolutionary fixed chromosomal rearrangement. For each EBR between loci 1 and 4, we found corresponding regions 1–2 and 3–4 based on a synteny map of the ancestral species; **b** cePC1 values plot showing compartmental states of loci 1–4 in the ancestral genome. Each line represents cePC1 values for four loci (1, 2, 3, and 4). Quantification of differences between pairs of loci: 1 vs. 4, 2 vs. 4, and 1 vs. 3 are shown in Supplementary Fig. 26; **c** Hi-C contact frequencies of each species compared to syntenic *An. coluzzii* Hi-C contacts (OBS). Control provided for each species (RND) was obtained by random shuffling of contact frequencies within each genomic distance bin. L upper index indicates that species liftovered to *An. coluzzii* genomic coordinates; **d.** Conservation of the 3D-structure of the synteny block between *An. coluzzii* and *An. albimanus* species separated by 100 millon years; **e** Comparison of Hi-C maps of syntenic loci of *An. coluzzii* and *An. stephensi*. Dark blue—conserved features, light blue—modified ones. Color bars on **d**, **e** panels reflect the contact frequency.

The first phase characterized by a U-shaped slope curve occurs between 10 kb and 1 Mb. For distances between 10 kb and ~200–500 kb slope decreases modestly starting from ~−0.8 and reaching the lowest point around −1, which is the characteristic value for the fractal globule, and then starts increasing. At ~1 Mb starts phase II, where the slope rapidly increases reaching values around −0.6 for genomic distances ~2–3 Mb and then falls sharply to the values below −1.

These two phases were observed in all *Anopheles* species (Fig. 10b and Supplementary Fig. 27A), and analysis of *Drosophila, Aedes*, and other insect's Hi-C datasets demonstrated very similar patterns (Fig. 10b and Supplementary Fig. 27B; see Supplementary Table 7 for list of all datasets used to produce Fig. 10). When we analyzed vertebrate Hi-C data, we again found the characteristic shape of the slope curve: the phase I, characterized by prominent decrease of the slope, was much more prolonged than in insects, and reaches a minimum at genomic distances ~800 kb for chicken and ~1–1.5 Mb for mammals; phase II displayed a maximum at ~6–8 Mb for chicken and ~15–20 Mb for mammals, and after this point we observed a sharp drop of the slope, similar to the insect curves (Fig. 10a, b and Supplementary Fig. 27C, D). Moreover, in metaphase chromosome Hi-C data we observed a slope curve similar to phase II, although the genomic distance corresponding to the maximum slope value was shifted ~3–4 Mb and slope increase was much stronger (Fig. 10b, yellow line).

We used available Hi-C datasets obtained from cells lacking cohesin subunit RAD21, cohesin release factor WAPBL or cohesin loader NIPBL to confirm that the U-shape of the slope curve observed in phase I reflects the formation of TADs, and that genomic distance characterized by the minimal slope value corresponds to the characteristic TAD length (see Fig. 10b and Supplementary Note II and associated supplementary figures). This allows us to provide estimations of TAD lengths without biases introduced by TAD-calling algorithms. The characteristic TAD length in *Anopheles* genomes is around 200–400 kb, which is very close to *Drosophila* species, and slightly smaller than *Aedes* TADs (500–800 kb).

We next aimed to explain changes of the slope during phase II, namely the abrupt drop of contact probabilities observed at genomic distance ~3 Mb for insects, ~8 Mb for chicken, and ~20 Mb for mammals. We confirmed that this slope decrease was seen in multiple experiments on unsynchronized cells, G2-phase synchronized chicken DT40 cells, G1-phase quiescent chicken erythroblasts and S-phase synchronized *Drosophila* S2 cells or embryonic cells (Supplementary Fig. 27A–D) cells, indicating that there is no cell type or cell cycle specificity and that observed results could not be explained by a fraction of mitotic cells. Moreover, cohesin-mediated or condensin-mediated loop-extrusion processes are not required to maintain chromosome conformation reflected by the drop of the slope, as we observed the drop under RAD21-degron conditions, NIPBL-degron conditions, both CapH-degron, CapH2-degron, and SMC2-degron conditions, and in vertebrate erythrocytes which lack extrusion-mediated TADs[47] (Fig. 10 and Supplementary

Fig. 27A–D). In addition, we observed phase II pattern in silkmoth data, although these animals lack CapH2 and CapG2 condensin II subunits[48].

We next speculated about possible constraints underlying the observed drop of the slope. We hypothesized that the decrease of contact frequencies observed at large distances might be explained by the specific shape of chromosome territory. Indeed, it is known that chromosome territories are not spherical in mammalian[49,50] and mosquito[51] cells. If DNA occupies an ellipsoidal or cylindrical shape, then the radius of the cylinder will determine the characteristic spherical volume inside which the drop of contacts should be in agreement with the fractal globule model. Upon exiting this territory, loci are located in different slices of the cylinder, which makes the likelihood of contacts between them smaller than expected for the fractal globule model. Following this hypothesis, our data suggest that the interphase chromosomes of all animals fill elongated volumes, characterized by the specific minor radius, different for insects, chicken, and mammals. We estimated that the size of a spherical globule of DNA that fits in this volume is ~1–3 Mb for insects, ~7 Mb for chicken, and ~20–25 Mb for mammals.

## Discussion

**Assembly of mosquito genomes using Hi-C data.** Originally developed to study 3D genome architecture, Hi-C is now widely applied to assemble various genomes[52,53]. Our results demonstrated that Hi-C-based scaffolding provides mosquito genome assemblies of high contiguity, especially when combined with long-read sequencing data as in the case with *An. merus*. Using Hi-C data, fixed and polymorphic inversions in *Anopheles* populations were successfully detected recently[32] and in our study. Accurate inversion breakpoint detection is important for understanding the mechanisms underlying speciation and adaptation of *Anopheles* mosquitoes[54]. The Hi-C method can be applied to other mosquitoes including species from genera *Aedes* and *Culex* to study their inversion polymorphisms.

While this work was in preparation, several *Anopheles* genome assemblies were generated using Hi-C and complementary approaches[36,38,55–57]. Further comparisons of these assemblies may reveal even strain-specific features of the linear and spatial genome organization.

**3-dimensional organization of *Anopheles* genomes.** We comprehensively profiled chromatin organization in *Anopheles* mosquitoes and showed how different chromatin features interfere with each other. We found that several properties of chromatin organization, identified previously in *Drosophila*, are shared by *Anopheles* species. In particular, *Anopheles* genomes are divided into distinct compartments, characterized by different gene density and expression levels, and that loci belonging to different compartments are insulated from each other by TAD boundaries, as previously shown by Rowley et al.[16] and reviewed in Lezcano et al.[58]. However, our data suggest that there are more than two types of compartments in the nucleus. For example, large

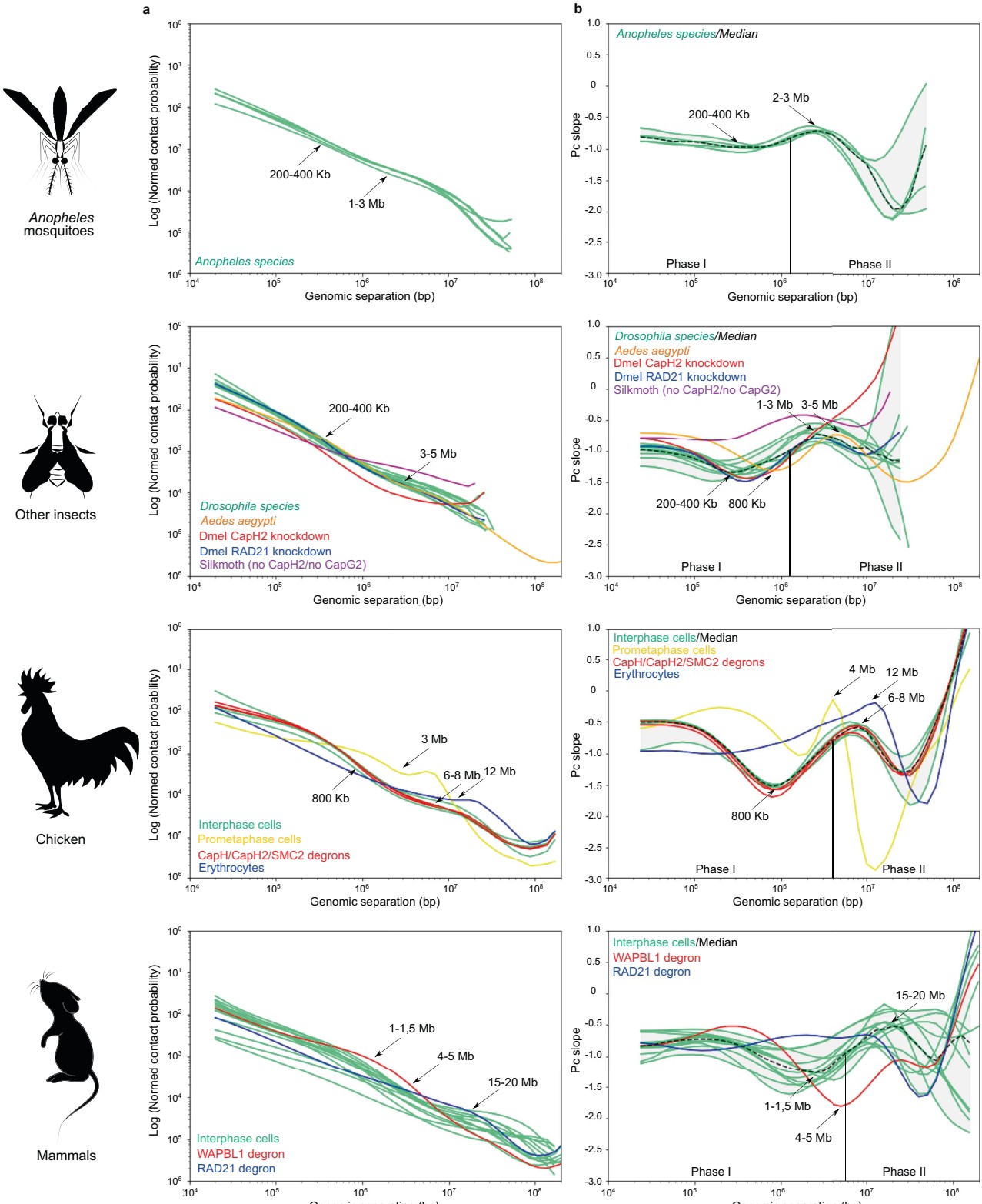

**Fig. 10 Contact frequency decays non-uniformly with genomic distance. a** Dependence of contact probability on genomic distance for *Anopheles*, other insects, chicken, and mammals shown in log–log coordinates; **b** Slope of the curve depicted in **a**, as a function of genomic distance. Black dashed line shows the median, and the gray area represents minimal and maximal values for several species. Slope-plots for individual species are shown as thin green lines.

intercalary heterochromatin blocks showed interaction patterns distinct from other B-compartment regions. Our work identified at least two types of B compartments in mosquitoes: euchromatic and heterochromatic. Large intercalary heterochromatin blocks,

cytogenetically defined as diffuse intercalary heterochromatin[42], form B-compartments with regions of pericentromeric heterochromatin, but they do not form B-compartments with euchromatic regions. Euchromatic regions form their own B-compartments,

similar to those described in *Drosophila*. Although *D. melanogaster* chromosomes do have compact intercalary heterochromatin, it has a different nature than diffuse intercalary heterochromatin found in *Anopheles*. Similarly, specific looping interactions were detected within H3K27me3-enriched regions, but not within other regions of the B-compartment. These data indicate that there are more than one type of B-compartments.

Complexity of mechanisms underlying genome architecture in mosquitoes could explain why a simple regression model based on gene density, GC-content, cePC1 values, and RNA-seq failed to accurately predict insulation scores across genomes. This can be either due to missing 1D information such as, for example, location of architectural proteins[59], or because more complex, nonlinear models should be developed to explain interdependencies between known 1D chromatin features. Some of these models are already developed to explain chromatin architecture in mammals[60,61] and can potentially be applied to *Anopheles* mosquitoes data.

Finally, we note that all experiments were performed on heterogeneous populations of embryo cells or adult mosquitos, which include multiple cell types. Thus, we can not discriminate cell type-specific properties of genome architecture; moreover, according to recent studies[15,62] certain chromatin structures observed at the level of cell populations may not exist in individual cells and, vice versa, chromatin nanodomain of individual cells may not be captured by standard Hi-C approach. Therefore, studying genome organization at the level of individual mosquito cells or pure cell population is an important direction of future research.

**Evolutionarily conserved long-range chromatin interactions in *Anopheles* genomes**. In all studied *Anopheles* species we found that certain genomic regions form giant loops spanning tens of megabases. Loops described previously in *Drosophila* Hi-C data[13] are typically smaller in size and mainly attributed to interactions between active genes or Polycomb proteins[63]. Polycomb-mediated Mb-scaled interactions have been shown to occur in mammalian cells[64,65].

We found three types of loops in *Anopheles* genomes. The first type, represented by mid-range interactions within H3K27me3-rich domains, corresponds to typical Polycomb-mediated loops described earlier in *Drosophila*[13]. The second type, represented by giant X-chromosome loops, shows moderate enrichment of H3K27me3 signal and, thus, could be also associated with Polycomb complexes, similar to the long-distance loops described previously in mammals[64,65]. However, these X-loops could not be fully explained by interactions of H3K27me3-rich regions, because other loci with similar or even higher H3K27me3 enrichment levels located at smaller genomic distances showed significantly less interactions. Here it is pertinent to note that Polycomb loop anchors have been shown to be bound by Pc subunit of the PRC1 complex[13], whereas H3K27me3 modification is deposed by the PRC2 complex[66]. Differential binding of PRC1 complexes could explain why some of the H3K27me3-rich regions are engaged in long-range looping whereas others are not.

Finally, we have shown that some giant loops are located outside of H3K27me3-rich regions, although their anchors are located in the B-compartment. The absence of H3K27me3-signal makes these loops different from long-range loops observed previously[64,65]. Formation of these loops could not be explained by currently known mechanisms. These loops are conserved for more than 100 million years, despite substantial reshuffling of *Anopheles* genomes. Large genomic distance between loop anchors and the presence of multiple insulation boundaries

located between them suggests that the loops are formed by chromatin clustering rather than extrusion-based mechanisms.

We noted that anchors of loops belonging to three different types did not interact with each other, indicating that there are at least three different molecular mechanisms underlying long-range looping interactions.

Evolutionary conservation of looping interactions implies its functional importance. However, FISH analysis showed cell-by-cell variability of contacts between loop anchors, suggesting that 100% formation of the loops is not essential for the function of follicle cells. If such variation exists in embryos then the Hi-C method is sensitive enough to identify the interaction in the mixed cells population. In the case of ovarian nurse cells, we found that colocalization correlates with the low-levels of chromosome polyteny. We conclude that the high polyteny likely creates physical properties of chromosomes that hinder specific interactions between the loop anchors. An alternative explanation is that highly-polytenized chromosomes do not require such interactions. Interestingly, no specific long-range interactions have been found in polytene chromosomes of *D. melanogaster* by microscopic analysis[67] or Hi-C[13].

**Chromosome territory shape constrains long-range contacts**. We have shown that long-range contacts are structurally constrained in genomes of all examined animal species, and suggested that chromosome territory shape might underline these constraints. It is not known what determines the non-spherical shape of chromosome territories. We confirmed that the drop of the long-range contact frequencies distances is still present after CapH2-knockdown in *Drosophila* cells, as well as in chicken cells with CapH, CapH2, or Smc2 proteins conditionally degraded. However, it is pertinent to note that chicken cells lacking condensin could not pass through mitosis, so that degradation of condensin subunits was induced in the G2-phase[68]. Thus, despite the fact that condensin proteins probably do not play a pivotal role in the maintenance of cylindrical chromosome territory shapes during interphase, we can not exclude that they are required to form these shapes during previous mitotic division. As mentioned above, RAD21 degradation also does not show any effect on the slope changes at large genomic distances, arguing against the role of cohesin in this process.

To explain our observations, we speculated that chromosomes acquire their elongated form during mitosis, and this elongated form is preserved during the cell cycle. We argue that maintenance of this mitotic-like shape does not require any specific mechanism, because time required for complete equilibration (or relaxation) of chromosomes after mitosis in *Drosophila* is ~500 years, which dramatically exceeds the lifetime of the entire organism[69]. Similar hypothesis was recently proposed to explain the difference between slope changes in mammalian oocytes arrested at the prophase stage for months and sperm cells, which are not subjected to such prolonged mitotic arrest[70].

This hypothesis suggests that the observed pattern of long-range contacts will be more pronounced in cells which have just exited the mitotic cycle or in cells where chromatin motion is more confined, for example due to a small nuclear size. And, indeed, our recent study showed that vertebrate erythrocytes, the cells with extremely compact nuclei which live for only a few weeks after exiting the mitotic cycle, displayed pronounced "second diagonal" contacts pattern[47]. In addition, Hi-C profiling of mitotic exit demonstrated the same "second diagonal" pattern in early G1-phase[71].

Another effect of the elongated chromosome territory shape is a reduction of the long-range compartmental interactions. At the

same time, we observed the clustering of centromeric and telomeric heterochromatin and increased frequency of trans-chromosomal interactions in the centromeric and telomeric regions. According to the principal component analysis, these factors dominate the genomic compartmentalization, although compartment interactions are present and well-pronounced at mid-range distances. We proposed an approach, "contrast enhancement", for the identification of compartments in genomes with a Rabl-like configuration and successfully employed it to define compartments in *Anopheles* genomes. This approach could be applied to other systems where compartmental interactions are attenuated at large distance, such as meiotic chromosomes[72] or plant genomes[45].

### Chromosomal evolution and 3-dimensional architecture of the genome.

The level of evolutionary conservation of 3-dimensional architecture of genomes is a subject of active debates. There is conflicting evidence either supporting[5,7] or disproving[19] the evolutionary stability of TADs in *Drosophila* species. The main argument of the former is the enriched frequency of EBRs near TAD boundaries. Our results obtained using *Anopheles* genomes confirmed the prevalence of EBRs in insulated regions; however, we also showed that both genomic insulation and EBRs tend to occur in gene-dense regions. Importantly, gene density alone could not fully explain the genomic insulation of TAD boundaries, suggesting that factors independent of gene density contribute to TADs formation. However, controls with gene density equal to EBRs distribution display almost the same level of insulation as observed for EBRs, defining gene density as a main factor for associations between genomic distribution of EBRs and TAD boundaries.

It is not clear why breaks of chromosomal rearrangements occur in gene-dense regions of *Anopheles* genomes. However, we noted that occurrence of EBRs in gene-dense regions was also shown in mammals and yeast[73]. One plausible explanation of this phenomenon could be related to the overlap between gene-dense regions and active chromatin[74]. In that case, breaks occur more often in the regions of increased chromatin accessibility, while misrepaired regions are brought into spatial contact by the three-dimensional conformation of chromosomes[73]. Indeed, our data showed that EBRs are significantly enriched in the A-compartment, and that the majority of rearrangements occurred between A-compartmental regions. Although this explanation connects 3D genome organization and distribution of EBRs, the connection does not imply evolutionary constraints caused by functional significance of TAD boundaries.

If there is no selective pressure maintaining TADs as intact units, why does TADs composition show remarkable similarity within syntenic blocks of species diverged ~100 million years ago? We argue that TADs reflect clustering of chromatin, and principles underlying this clustering, such as Polycomb-mediated interactions[13], histone interactions[18,75], and organization of other nuclear condensates[76] are conserved across species. Chromatin properties, such as histone modifications are also highly conserved within syntenic blocks[77]. Thus, colinearity of genomic segments with conserved epigenetic status results in similarity of 3-dimensional chromatin organization, including TADs. As long as collinearity and epigenetic status remain conserved, the TADs organization will remain conserved as well.

In addition to the absence of evolutionary constraints on TADs architecture, we found that selection of a translocation partner does not depend on the ancestral epigenetic states of the rearranged loci, except the fact that all breakpoints are significantly enriched in the A-compartment. Why is there no preference for joining the loci with similar epigenetic status? We

consider that majority of genes are regulated by local cis-elements located at a small distance from gene. These relatively autonomous units represent micro-TADs, which at the resolution of our analysis (5 kb) are often encompassed within one or two bins and, thus, are not detected. When a continuous region includes several units with the same epigenetic status, their interactions result in formation of TAD. However, each unit is largely autonomous, and when it is translocated to a new epigenetic environment, it could establish local or distant interactions with other units characterized by the same epigenetic state. The existence of gene-scale micro-TADs, which are autonomously regulated, is supported by recent high-resolution analyses of the *Drosophila* genome architecture[3,16] and by the analysis of 3-dimensional organization and evolution of long genes[7].

Even with the advent of new, high-resolution Hi-C methods such as DNAseI[78] and MNAse[79], Hi-C detection of micro-TADs is challenging because, when sharing epigenetic status and located continuously, they are not insulated from each other. Evolutionary comparisons, such as identification of genome regulatory blocks[80], may help to dissect these basic units of chromatin organization in future.

## Methods

**Mosquito colony maintenance.** Laboratory colonies of the following strains were used for the experiments: MOPTI strain of *An. coluzzii* (MRA-763); MAF strain of *An. merus* (MRA-1156); Indian strain of *An. stephensi* (Virginia Tech); EBRO strain of *An. atroparvus* (MRA-493); STECLA strain of *An. albimanus* (MRA-126). The mosquito strains (except *An. stephensi*) were initially obtained through Malaria Research and Reference Reagent Resource Center (MR4) stocks and BEI Resources, NIAID, NIH (Anopheles program; [https://www.beiresources.org/AnophelesProgram/Anopheles/WildStocks.aspx]). *An. coluzzii* Ngousou colony was created in 2006 from the broods of wild-caught pure *An. coluzzii* females in Cameroon[24], MOPTI strain of *An. coluzzii* was established by Professor G. C. Lanzaro from wild-caught Anopheles coluzzii females collected from Mali.

All colonies were maintained in the insectary of the Fralin Life Science Institute at Virginia Polytechnic Institute and State University. Mosquito specimens were hatched from eggs in unsalted water and incubated for 10–15 days undergoing four larvae and pupae developmental stages at 27 °C. Adult mosquitoes were maintained in the incubator at 27 °C, 75% humidity, with a 12 h cycle of light and darkness. Five to seven days of adult mosquitoes were blood-fed on defibrinated sheep blood using artificial bloodfeeders. Approximately 48–72 h of post-blood feeding, the egg dishes (Supplementary Fig. 28) were placed and after 15–18 h embryos were collected for further experiments.

**In situ Hi-C on mosquito embryos and adults.** The step-by-step mosquito Hi-C protocol can be found in Supplementary Methods. In brief, procedure for mosquito embryos was modified based on the previously published high-resolution 3C protocol[81], *Drosophila* Hi-C protocol[63], and in situ Hi-C protocol for mammalian cells[1]. Mosquito eggs were collected (optimized egg dish can be found in Supplementary Fig. 27) and Hi-C libraries were prepared using nuclei isolated from ~1000 to 3000 embryos of mixed sexes. Embryos were fixed with 3% formaldehyde at the developmental stage of 15–18 h after oviposition. MboI restriction enzyme (NEB, #R0147) with average restriction fragment size ~250–300 bp was used in the experiment. Two biological replicates of Hi-C libraries were generated, prepared with NEBNext® Ultra™ II DNA Library Prep Kit for Illumina (NEB, #E7103), and sequenced using 150 bp pair-ended sequencing on Illumina platform. For Hi-C library preparation from *An. merus* adult mosquitoes, 15 males in two biological replicates were homogenized with liquid nitrogen in a precooled mortar. Separate fixations were done for each Hi-C library of *An. merus* adult males and the downstream procedures were the same as for Hi-C libraries from embryos.

**PacBio sequencing.** About 100 one-to-two-day old adult males of *An. merus* generated from a single pair of grandparents were used for extraction of the whole genomic DNA. Adult mosquitoes were ground to a fine powder with liquid nitrogen in a precooled mortar. High-molecular weight DNA was extracted using Blood & Cell Culture DNA Midi Kit (Qiagen, Hilden, Germany) with 100/G tips. Extracted DNA was purified with Genomic DNA Clean & Concentrator −10 kit (Zymo Research, Irvine, CA, USA). A total of 17.9 µg of high-molecular weight DNA was used for PacBio sequencing. The DNA concentration of 179 ng/µl in 100 µl was measured by the Qubit Fluorometer (Invitrogen, Carlsbad, CA, USA). PacBio sequencing was performed at the Duke Center for Genomic and Computational Biology (Duke University, Durham, NC, USA). Detailed sequencing reads statistics can be found in Supplementary Data 2.

**ChIP-seq**. The anti-trimethyl-histone H3 (Lys27) (Millipore, #07−449) antibody has been successfully validated by Western Blot (Supplementary Fig. 29) and then used for immunoprecipitation experiment. We followed the protocol previously published in Akulenko et al.[82] with some modifications. Briefly, ~1000–2000 eggs were bleached, homogenized, and fixed according to optimized ChIP-seq protocol (Supplementary Methods). Cells were lysed and chromatin was sonicated using Bioruptor Diagenode machine, 8–10 cycles of 10/10 s ON/OFF in the lysis buffer (140 mM NaCl, 15 mM HEPES, 1 mM EDTA, 0.5 mM EGTA, 1% triton, 0.5 mM DTT, 0.1% sodium deoxycholate, protease inhibitors) with SDS and N-lauroylsarcosine. After determining the total concentration of chromatin by Qubit, 5–10 µg were used per one ChIP reaction. Additionally, we used *D. melanogaster* chromatin as a spike-in (5–10% of total chromatin amount). Before incubation with antibody, chromatin was pre-cleared by incubation with agarose beads (Pierce™ Protein A/G Agarose, ThermoFisher #20421) for 2 h at 4 °C with slow rotation. During this time, another aliquot of agarose beads was washed, blocked with BSA, combined with target antibody, and incubated overnight at 4 °C with slow rotation. After washing step, pre-cleared chromatin was immunoprecipitated with antibody-agarose beads complexes and incubated overnight at 4 °C with slow rotation. Next day, the beads were thoroughly washed in a series of buffers (LB: 150 mM NaCl, 20 mM Tris-HCl, 2 mM EDTA, 1% triton, 0.1% SDS; HB: 500 mM NaCl, 20 mM Tris-HCl, 2 mM EDTA, 1% triton, 0.1% SDS; LiB: 0.25 M LiCl, 10 mM Tris-HCl, 1 mM EDTA, 1% NP-40; TE buffer). DNA was eluted with 250 µL elution buffer (EB: 1% SDS, 0.1 mM NaHCO₃) by incubation at 65 °C for 10 min. To revert cross-linking, the DNA was incubated at 65 °C overnight in presence of 0.25 M NaCl, 10 mM EDTA and 40 mM TrisHCl. DNA was extracted using phenol–chloroform mix followed by ethanol precipitation. ChIP-seq libraries were prepared for sequencing using NEBNext® Ultra™ II DNA Library Prep Kit for Illumina (NEB, #E7103). The detailed ChIP-seq protocol can be found in Supplementary materials.

**RNA-seq**. One thousand five hundred to two thousand embryos of 15–18 h *Anopheles* mosquitoes were bleached for 5 min. Total RNA was extracted following the Monarch Total RNA Miniprep Kit (NEB #T2010S) protocol with minor modifications. Mosquito embryos were homogenized in 800 µL of lysis buffer with 2 mL Dounce homogenizer. Samples were incubated at RT for 10 min, then proteinase K was added and samples were incubated for an additional 5 min at 55 °C. After that, the tubes were centrifuged at max speed for 2 min. Supernatant was transferred to fresh RNAse/DNAse-free tubes and proceeded with gDNA removal columns. The incubation time with DNAse was increased to 20 min in total. Total RNA was eluted with 50 µL H2O. Sample concentration was verified with Nanodrop and 1 µg of total RNA was used for the next procedures. Samples were prepared for Illumina sequencing with NEBNext® Ultra™ II RNA Library Prep Kit for Illumina (NEB #E7775) accompanied by NEBNext Poly(A) mRNA Magnetic Isolation Module (NEB #E7490) with RNA insert size of 200 bp.

**Ovary preservation and polytene chromosome preparation**. To prepare high-quality polytene chromosome slides, we followed the protocol described previously[83] with minor exceptions. Approximately 24–30 h after the second or third blood feeding (the timeline for Christopher's III developmental stage varies for different species and should be estimated by visual inspection), ovaries were fixed in Carnoy's solution (3:1, ethanol:glacial acetic acid by volume), kept at RT for 24 h, then stored at −20 °C for a long term.

At least 1 week after fixation, ovaries were dissected in Carnoy' solution. Cleaned and separated follicles were then placed on a slide in a drop of 50% propionic acid for ~5 min (5–10 follicles per slide), where they were macerated and squashed in a fresh portion of 50% propionic acid. The quality and banding pattern were briefly examined using a phase-contrast microscope (1000×) and high-quality preparations were proceeded with snap-freezing in liquid nitrogen. After freezing the slides were immediately placed in pre-chilled 50% ethanol and kept at −20 °C overnight. Next day, after removing coverslips, preparations were dehydrated in ethanol series (50, 70, and 96%), air-dried, and the quality of polytene chromosomes was checked. High-quality slides were placed in a cardboard holder and stored at RT up to 3 months.

**2D-FISH**. Probes were prepared by the Random Primer Labeling method described in Protocols for Cytogenetic Mapping of Arthropod Genomes[84]. FISH probe sizes varied between 500–1500 bp and were mainly designed for exon regions. gDNA was freshly extracted using Monach Genomic DNA purification kit, PCR product amplification was performed by regular PCR with DreamTaq/Q5 polymerase. PCR Product was purified with Qagen purification columns and ~200–250 ng were used for 25 µL labeling reaction. After overnight incubation in the thermocycler at 37 °C, labeled probes were precipitated with 96% ethanol, dried, and dissolved in 30 µL hybridization buffer. Ten to fifteen microliter of one probe were applied to slide. Prepared slides were incubated for 25 min at 70 °C in a humid chamber following the overnight incubation at 39 °C. Washing steps included two times wash in 1× SSC at 39 for 20 min, one time wash in 1× SSC at RT for 20 min, one time wash in 1× PBS at RT for 10 min. One drop of ProLong™ Gold Antifade Mountant with DAPI (ThermoFisher #P36931) was added to the slide for protection. Fluorescent signals were detected and recorded with a Zeiss LSM 710 laser scanning

microscope (Carl Zeiss Microimaging GmbH, Oberkochen, Germany). Set of primers for PCR product amplification can be found in Supplementary Data 9.

**3D-FISH**. Fluorescent probes were prepared by the same method as for 2D-FISH. Probe pellets were dried and resuspended in 20–30 hybridization buffer. Twenty microliter of one probe were used per one experiment where the total volume of the hybridization probe solution contained at least 80 µL. Probes were denatured in Thermomixer at 90 °C for 5 min, then transferred to 39 °C and pre-annealed for at least 30 min.

Twenty-four to thirty-hours hours after blood feeding ovaries were dissected in 1× PBS and placed in 1 mL 1× PBST. Fixation was performed in 4% paraformaldehyde solution for 20 min with rotation. Then, ovaries were washed in PBS and treated with 0.2 µg/µL RNAse solution in PBS at 37 °C for 20 min. After rinsing in PBS, ovaries were placed in 1% Triton-X-100/0.1 M HCl solution and incubated at RT for 20 min with rotation. After brief washing, DNA denaturation was performed in 50% formamide/2 × SSC solution at 75 °C for exactly 30 min. Then, ovaries were rinsed in 100 µL of hybridization buffer which was replaced with hybridization mix. Tubes were incubated at 39 °C overnight with slow mixing. Next day, the ovaries were washed in a series of washing solutions at 39 °C with rotation: three washes in 50% formamide/2 × SSC; three washes in 2 × SSC. Drop of ProLong™ Gold Antifade Mountant with DAPI was added and ovaries were placed on a 3D-FISH slide. Fluorescent signals were detected and recorded using a Zeiss LSM 880 confocal laser scanning microscope (Carl Zeiss AG, Oberkochen, Germany).

### Computational methods

*Hi-C data processing*. Raw reads were processed using Juicer protocol[85]. Contacts were normalized using KR-normalization. Expected contact counts were obtained by dumping expected vectors using juicer tools *dump* tool.

For insulation score and contacts scaling computation data was converted to cool format using hic2cool convert function with default parameters. The insulation score values were normalized for by subtracting a mean score within +/− 1 Mb frame around the target bin.

*Genome assembly*. 3D-DNA pipeline[4] (version 170123) was employed to assemble the genomes de novo using the generated Hi-C data set. Misassemblies were identified and fixed manually using assembling mode in Juicebox software[85,86]. The physical genome maps were used to assess the assemblies[22,23,27–29].

*Whole-genome alignments, synteny blocks and CE calling*. For pairwise whole-genome alignments we used LastZ tool[87] with parameters: high-scoring segment pairs (HSPs) threshold (-hspthresh) = 6000, interpolation threshold (-inner) = 2000, step size (-step) = 20, alignment processed with gap-free extension of seeds, gaps extension of HSPs and excluded chaining if HSPs (-gfextend -nochain -gapped). The generated.maf-files were converted to.net-format with KentUtils [https://github.com/ENCODE-DCC/kentUtils].

We initially defined synteny blocks as alignment blocks reported in.net-file. Next, we iteratively merged blocks, using the following strategy. Let A1 and A2 represent two consecutive synteny blocks found in species A, and B1 and B2 represent corresponding synteny regions for species B. We merged A1 and A2 into A12 if 1) orientation of both blocks A1 and A2 are the same; 2) orientation of both blocks B1 and B2 are the same; 3) the lengths of gaps between synteny blocks in both species were less than 150 Kb; 4) the difference in length between the gaps identified in species A (gap A1–A2) and species B (gap B1–B2) was less than 100 Kb.

We iteratively performed a merging procedure until no changes could be made, and then filtered out all synteny blocks less than 15 Kb in length.

To find multi-species alignment blocks we used Mugsy[88] with default parameters. To call CE, we used PhastCons. Parameter tunings were guided by a software recommendation, but 65% exon coverage by CE was an unreachable goal. To solve this problem, we analysed a relation between a summarized length of CEs and the parameters, when the phylogenetic information threshold was near ten bits. We observed a "plateau" in increasing summarized CEs length when target coverage and expected coverage of CE varied between 0.50 to 0.60 and from 40 to 50 nucleotides, respectively. Thus, for each alignment we fixed parameters at the point when summarized CEs length reaches plateau.

*Phylogenomic analysis and calculation of divergence times*. We analysed genome assemblies of the six mosquito species available from VectorBase[89] release VB-2019-08 with the Diptera dataset of the Benchmarking Universal Single-Copy Orthologue (BUSCO v3.0.2) assessment tool[90]. From the results, we identified 1258 BUSCO genes present as single-copy orthologues in all six species. We aligned the protein sequences for each BUSCO with MAFFT v7.450[91] and then filtered/ trimmed them with TrimAl v1.2[92] using automated parameters to produce a concatenated superalignment. AliStat v1.12[93] assessment of the superalignment: six sequences; 854,431 sites; completeness score: 0.96383. We then estimated the phylogeny using RAxML v8.0.0[94] with the PROTGAMMAJTT model with 100 bootstrap samples. Rooted with *Aedes aegypti*, we converted the molecular phylogeny to an ultrametric time-calibrated phylogeny using the chronos function in R[95] with the discrete model and fixing the *An. gambiae* complex age at 0.5 million years according to Thawornwattana et al.[21] and the *Anopheles* genus age at 100

million years in line with Neafsey et al.[20] and the geological split of western Gondwana.

*ChIP-seq data processing.* All ChIP-seq data were processed using the Aquas pipeline [https://github.com/kundajelab/chipseq_pipeline] stopped at the signal stage.

*RNA-seq data processing.* All data were processed using standard protocols with HISAT2, deeptools bamCoverage, StringTie tools[96]. The sequencing data were uploaded to the Galaxy web platform, and we used the public server at usegalaxy.org to analyze the data[96]. We used averaged TPM values obtained from three biological replicas for all downstream analysis.

*Compartment calling.* The default approach for PC1 value computation relied on using juicer tools *eigenvector* tool[85] at 25, 50 or 100 kb resolutions. For other approaches, explained in details in Supplementary Note I, we used custom R-scripts [https://github.com/labdevgen/ABCE]. To compute PC1 values for intrachromosomal submatrices we used window (frame) size equal to 10 Mb.

*TAD calling.* To call TADs, we first used Armatus, Lavarbust, Dixon callers and hicExplorer *findTADs* utils[59,97–99] with default parameters at 25 and 5 kb resolutions. Visual inspection of obtained TADs revealed that results were similar for the Armatus, Lavarbust, and hicExplorer algorithms, whereas the Dixon caller resulted in large, Mb-scaled TADs, which did not correspond well with triangles visible on contact maps. Although this difference is most probably due to default parameters of the Dixon TAD caller, originally developed on mouse and human data, and results could be improved by tweaking parameters, we decided to focus on hicExplorer-based TADs at 5 kb resolution because visual assessment suggested that this caller provided the best results.

By tweaking parameters as suggested in the manual, we found that a delta value of 0.05 provided TADs most closely matching triangles on contact maps; changing other parameters did not lead to substantial improvements of TADs. Finally, resulting TADs were visually inspected to correct boundary positions in problematic regions (long heterochromatin blocks, gaps, and etc).

*Slope plot analysis.* To produce slope plots, we computed the contact frequency (P) as a function of genomic distance (s), as well as corresponding derivative values using cooltools expected function (https://github.com/mimakaev/cooltools/blob/master/cooltools/). Resolution in ten bins per order magnitude was used for plots.

**Statistics and reproducibility**. Hi-C experiments were repeated in two independent replicas for all *Anopheles* species. Inversion loops for *An.stephensi* (known 2Rb) and *An.atroparvus* (new 2L1) were detected within 2–6 individual polytene chromosome slides. FISH experiments were reproducible. The complete statistics for 3D FISH experiments shown in Fig. 4c and Suppl. Fig. 11g can be found in Suppl. Table 13.

**Reporting summary**. Further information on research design is available in the Nature Research Reporting Summary linked to this article.

## Data availability

All data generated or analyzed during this study are included in this published article and its Supplementary files. The raw sequencing data for five *Anopheles* species have been deposited in the NCBI SRA database with the following accession numbers: PRJNA615788 (RNA-seq raw reads), PRJNA623252 (ChIP-seq raw reads), PRJNA615337 (Hi-C raw reads for embryos), and PRJNA630123 (Hi-C raw reads for *An. merus* adults). Processed data, including genome assemblies, Hi-C contact maps, RNA-seq and ChIP-seq tracks, TADs and compartments are available at https://genedev.bionet.nsc.ru/Anopheles.html. Genome assemblies are also available at NCBI (BioProject: PRJNA660041, genome accessions: JADFFJ000000000, JADGIR000000000, JADGIQ000000000, JADFFN000000000, JADFFO000000000). Note that the NCBI references are slightly different from those deposed at https://genedev.bionet.nsc.ru/Anopheles.html due to contamination filters applied by the NCBI team. Source code for the ABCE tool computing cePC1 values is available here: https://github.com/labdevgen/ABCE. Miscellaneous scripts are available at [https://github.com/labdevgen/Anopheles_Ps]; [https://github.com/labdevgen/lavaburst_domains]; [https://github.com/labdevgen/ANopheles_Rabl].

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

## Acknowledgements

The following reagents were obtained through BEI Resources, NIAID, NIH: *An. coluzzii*, Strain MOPTI, Eggs, MRA-763, contributed by Gregory C. Lanzaro; *An. merus*, Strain MAF, MRA-1156, contributed by Maureen Coetzee; *An. atroparvus*, Strain EBRO, Eggs, MRA-493, contributed by Carlos Aranda and Mark Q. Benedict; *An. albimanus*, Strain STECLA, Eggs, MRA-126, contributed by Mark Q. Benedict. All computations were performed using nodes of the high-throughput cluster of the Novosibirsk State University, and bioinformatics resource center of the Institute of Cytology and Genetics. We are sincerely grateful to Nariman Battulin for fruitful discussions. This work was supported by the NSF grant MCB-1715207, NIH NIAID grants R21AI135298 and R21AI159382, and the USDA National Institute of Food and Agriculture Hatch project 223822 to IVS. The reported study of *An. atroparvus* was partly funded by RFBR according to the research project no 19-34-50051 to IVS and VL. PacBio sequencing of *An. merus* was funded by a grant from the University of Lausanne Department of Ecology and Evolution to RMW and NIH grants AI133571 and AI121284 to Z.T. M.J.M.F.R., L.R., R.F., and R.M.W. were supported by Novartis Foundation for medical-biological research grant #18B116 and Swiss National Science Foundation grants PP00P3_170664 and PP00P3_202669. V.L. was partly supported by the Fulbright Foreign Student Program, Grantee ID: 15161026. Hi-C data analysis was supported by the Ministry of Education and Science of Russian Federation, grant #2019-0546 (FSUS-2020-0040). ChIP-seq data analysis was supported by project 121031800061-7 (Mechanisms of genetic control of development, physiological processes and behavior in animals).

## Author contributions

V.F. and I.V.S. conceived and supervised the study. V.L. performed all Hi-C, ChIP-seq, RNA-seq and FISH experiments, with help from J.L. in Hi-C-experiments on *An. merus* adult mosquitoes. M.N. developed the ABCE tool and performed most of Hi-C data analysis with help from A.T. P.B. performed ChIP-seq and RNA-seq data analysis. M.N. and V.L. generated genome assemblies and analyzed genomic inversions. J.L., Y.W., C.M., Z.T., M.J.M.F.R., L.R., and R.M.W. generated and analyzed PacBio data. R.F. and R.M.W. performed BUSCO analysis. All the authors contributed to the manuscript preparation.

## Competing interests

The authors declare no competing interests.
