## [Peer Review File · Nature Communications]

Anopheles mosquitoes reveal new principles of 3D genome organization in insectsReviewers' Comments:

Reviewer #1:

Remarks to the Author:

The MS by Lukyanchikova et al describes a comprehensive analysis of the genome spatial structure in *Anopheles* mosquitoes and present Hi-C-based assemblies of their genomes. Overall, this is a well-performed study which would be of interest for specialists in conventional and 3D genomics, and thus is suitable for publication in Nature Communications.

I have several minor concerns to be addressed prior publication:

1. One impressive finding is the presence of extremely large evolutionary conserved loops. Referring to Eagen 2017, the authors tried to check if these loops are Polycomb-dependent. However, ChIP-seq with anti-H3K27me3 antibodies is not the best choice because Eagen with co-authors showed that loops are formed not between H3K27me3-occupied regions (which are typically relatively long), but between punctate binding sites of Polycomb proteins (Pc, in particular). Thus, the authors should perform ChIP-seq with anti-Pc antibodies.
2. In Abstract in the sentence "However, in-depth analysis showed a confounding effect of gene density on both insulation and distribution of synteny breakpoints, suggesting limited causal relationship breakpoints and regions with increased genomic insulation" a word "between" is missed.
3. Table1: for readability, it would be better to change the column width to fit the content.
4. In the Intro section, the authors state: "...in contrast to mammals, the process of loop extrusion does not define the structure of chromatin contacts (Rowley et al. 2017; Kaushal et al. 2021)." This statement sounds too strong. Since the cohesin complex is the driver of extrusion, one need Rad21/Smc3 KO or induced degradation to check its role in the insect 3D genome. As far as I know, no such experiments have been reported to date. Moreover, Kaushal et al. showed that CTCF-KO results in the loss of some well-defined domain boundaries (see Fig. 5 in the original paper).
5. Hi-C maps should be supplied with a color bar.
6. The statement "...the PC1 and PC2 values depend on genomic distance, too" sounds a bit confusing. In this particular case this suggests the dependence on the locus-telomere distance. I feel the authors should rephrase this in some way to make this statement less general.
7. The authors should briefly explain the terms "Framed PC1" and "Cropped PC1" (fig. 4a,b) in the Results and reasons for the calculation of these profiles.

Reviewer #2:

Remarks to the Author:

In the manuscript, NCOMMS-21-18764-T, Lukyanchikova et al generated Hi-C datasets in five *Anopheles* mosquito species (with embryos or adult mosquito bodies). Firstly, using these datasets, the authors were able to improve existing genome assemblies of three mosquito species and successfully performed de novo assembly for two others, and identified large polymorphic chromosomal rearrangements in these mosquito genomes. Secondly, the Hi-C data revealed the known architectural features of 3D genome organization that have been identified in other species,

including the Rabl conformation, A/B compartments, TADs, and Chromatin loops, etc. The authors were able to identify several evolutionally- and developmentally conserved giant chromatin loops that span very large genomic distance (tens of megabases). Lastly, the authors performed evolutionary analysis and found that chromatin architecture within syntenic blocks remains remarkably stable between species.

Major concerns

1. The genome size and ploidy. Are the genomes of these mosquito species diploid or polyploidy?
2. The authors said that they performed the Hi-C assays in each sample with biological replicates. However, they did not include any result proving the quality of reproducibility.
3. A whole embryo or adult body contains different types (different lineages/tissues/organs) of cells and the cell populations are highly heterogeneous. Hence, the observed architectural features of 3D genome organization in each species are averaged result of the different cell types --- For those features that are largely conserved among different cell lineage/tissue types, such as Rabl configuration, A/B compartment and TAD organization, this will not be a big problem. However, for the fine-scale features that are more cell type-specific, such as chromatin loops, this will cause problems. A good example for this is the discrepancy regarding the giant loops between Hi-C and the 3D FISH results. From the Hi-C data, the authors said that these long-range loops are both evolutionarily and developmentally conserved, whereas in 3D FISH assays they appeared cell type-specific. To clarify this discrepancy, high-resolution 3C/4C assays in purified cell population might be a good choice.
4. Due to the heterogeneity issue described in the above point, the contact frequency decay curves in Fig 10 might not truly reflect what occurs in each cell type in each mosquitoes species.
5. In the title the authors claimed Anopheles mosquitoes reveal new principles of 3D genome organization in insects, which seems not quite true, i.e., the data doesn't show much new principles of 3D genome organization in insects.

Minor concerns

1. Table S5 meant all the giant loops?
2. What are the size of the FISH probes used?
3. It seems the giant loops are just structural features without functional implications.

Reviewer #3:

Remarks to the Author:

The authors extensively characterize the genomes and genome organizations of five different Anopheles species using Hi-C and other data. The work is an important contribution to the field both as a resource and in terms of the results revealed with respect to the conservation of genome organization across these species and how this relates to other organisms. Also, an additional contribution of the paper is the introduction of a new way to do chromatin compartment analysis (contrast enrichment) when regular PC analysis does not accurately capture active vs inactive compartments, which is generally the case for chromosomes folded in Rabl configuration across many

different organisms. However, the paper can certainly benefit from a bit of streamlining and restructuring to make sure the main points come across. I also have additional comments that need to be addressed before publication.

1. Since the Hi-C-based assemblies they have put together are done through manual annotation, it is imperative to further quality check these using read-level information around breakpoints to convincingly show that these are assemblies that can be used by others. The authors already indicate some concerns about Hi-C assemblies, as well as PacBio scaffolding, when they mention AgamP4.
2. The suggestion that the B compartment has at least two different forms can be followed up using subcompartment analysis as was done before for mammalian genomes.
3. The authors should discuss the potential relationship of the giant loops that are located outside of H3K27me3-rich regions to DNA methylation nadirs described by Goodell lab in 2020.
4. They need to elaborate on "confirmed that all these cases were due to errors in TAD or cePC1 calling". Were these mainly issues in compartment or TAD calling? I suspect mainly due to TAD calls. They need to show aggregate plots around such problematic calls to show that cePC1 values do not behave similarly when they are around non-problematic boundaries.
5. For X-loop and A-loop, and for other similar extremely long ones, sufficiently detailed analysis at the read-level is needed to eliminate the possibility that they correspond to structural variation at the DNA level rather than loops. I am not a FISH expert; therefore, I cannot be sure whether the differences between low and high polytene chromosomes rule out this possibility.
6. Extremely long-distance loops are presented and listed in supplementary, however, other loops have not been characterized in much detail beyond the H3K27me3 association.

- For each Hi-C heatmap, the authors need to indicate the resolution used (10kb? 25kb?) and a color scale.
- conservative features -> conserved features
- H3K27me3-reach -> H3K27me3-rich
- "active A-compartment (gene dense, expressed, GC-rich)": as their correlation tables in the supplement indicate, for most of the species, the cePC1 values did not correlate with GC content. Therefore the sentence containing this part needs to reflect the data.

Response to the reviewer's comments

We are deeply grateful to the Reviewers for taking their time to provide valuable comments and suggestions, which help us to improve the manuscript. Below, we provide our responses (in bold text) point by point to each comment. Direct citations of the revised manuscript are quoted. We note that we did not include some new data presented in this document in the manuscript text because this peer-review file will be included as supplementary information according to the journal's rules.

REVIEWER

COMMENTS

Reviewer #1 (Remarks to the Author):

The MS by Lukyanchikova et al describes a comprehensive analysis of the genome spatial structure in Anopheles mosquitoes and present Hi-C-based assemblies of their genomes. Overall, this is a well-performed study which would be of interest for specialists in conventional and 3D genomics, and thus is suitable for publication in Nature Communications. I have several minor concerns to be addressed prior publication:

1. One impressive finding is the presence of extremely large evolutionary conserved loops. Referring to Eagen 2017, the authors tried to check if these loops are Polycomb-dependent. However, ChIP-seq with anti-H3K27me3 antibodies is not the best choice because Eagen with co-authors showed that loops are formed not between H3K27me3-occupied regions (which are typically relatively long), but between punctate binding sites of Polycomb proteins (Pc, in particular). Thus, the authors should perform ChIP-seq with anti-Pc antibodies.

We completely agree with Reviewer#1 recommendation to use more specific Polycomb proteins (Pc or Ph) antibodies in addition to the H3K27me3 histone mark to prove the long-distance loop independence of the Polycomb complex. However, performing Pc or Ph ChIP-seq is complicated by the absence of commercial or custom mosquito antibodies. As a generous gift from Prof. Cavalli, we have received a Ph-antibody which was originally produced and verified for Drosophila cells (Loubiere et al. 2016; Loubiere et al. 2020). So, before proceeding with ChIP-seq, we have validated the antibodies in immunostaining experiments. First, we tested Ph-antibodies for Drosophila cell line S2

and it worked well (see Additional Fig.1, A). But based on our immunostaining results for the *An. stephensi* cell line MSQ43, the Ph-antibodies did not work for mosquitoes at all in our both attempts (see Additional Fig.1, B-C).

Additional Figure 1.

A.

0.1% triton/PBS, 3% BSA/PBS, Pr 1:200, Sec Gt488 1:300

B.

0.1% triton/PBS, 3% BSA/PBS, Pr 1:200, Sec Gt488 1:500

C.

0.1% triton/PBS, 5% BSA/PBS, Pr 1:200, Sec Gt546 1:500

It should be noted that immunostaining for various histone modifications worked nicely for the same cell line. Therefore, we assume that the Ph-antibody produced for *Drosophila* is not suitable for mosquitoes. Sequence alignment also showed a high divergence between Ph-subunit from *Drosophila* and *Anopheles*.

In addition, we note that although Eagen et al. showed that loop anchors preferentially overlap with Pc peaks, they also showed that these peaks are located within H3K27me3-rich areas: “*Loops were readily identified within H3K27me3-enriched regions*” (Eagen et al. 2017: Figure 2A and B and Figure 3 A and B). Our H3K27me3-data shows either moderate signal (for some of X-loops) or absence of signal (for A-loops). Other studies also showed a large (Schuettengruber et al. 2009, Fig. 2A), although not complete (Loubiere et al. 2016) overlap between H3K27me3 domains and Pc-peaks. Thus, we believe that PRC1 binding could not fully explain the formation of X-loops.

2. In Abstract in the sentence “However, in-depth analysis showed a confounding effect of gene density on both insulation and distribution of synteny breakpoints, suggesting limited causal relationship breakpoints and regions with increased genomic insulation” a word “between” is missed.

Thank you for pointing out the typo, it has been corrected.

3. Table1: for readability, it would be better to change the column width to fit the content.

Thank you for that recommendation, we will follow it and change the table 1 configuration according to the journal/technical editor instructions as soon as the manuscript will go through the final proofs.

4. In the Intro section, the authors state: “...in contrast to mammals, the process of loop extrusion does not define the structure of chromatin contacts (Rowley et al. 2017; Kaushal et al. 2021).” This statement sounds too strong. Since the cohesin complex is the driver of extrusion, one need Rad21/Smc3 KO or induced degradation to check its role in the insect 3D genome. As far as I know, no such experiments have been reported to date. Moreover, Kaushal et al. showed that CTCF-KO results in the loss of some well-defined domain boundaries (see Fig. 5 in

the

original

paper).

We agree with this point, and we modified the sentence as follows:

“These studies suggested that, in contrast to mammals, CTCF-mediated insulation plays only a limited role in the organization of chromatin contacts in Drosophila genome (Rowley et al. 2017; Kaushal et al. 2021).”

5. Hi-C maps should be supplied with a color bar.

We have now supplied all Hi-C heatmaps in Main figures and Supplementary materials with a color bar and resolution indicator.

6. The statement “...the PC1 and PC2 values depend on genomic distance, too” sounds a bit confusing. In this particular case this suggests the dependence on the locus-telomere distance. I feel the authors should rephrase this in some way to make this statement less general.

We have rephrased the paragraph as follows:

“Analysis of PC1 and PC2 distributions (Fig. 5, C) showed a pattern very similar to those observed previously in barley (Mascher et al. 2017), a plant species with Rab1-like configuration of chromosomes. Similar to mosquitoes, PC1 and PC2 values in plants reflect the position of the locus on the centromere-telomere axis. This additionally supports the link between Rab1-like chromosome configuration and specific distribution of PC1/PC2 values.”

7. The authors should briefly explain the terms “Framed PC1” and “Cropped PC1” (fig. 4a,b) in the Results and reasons for the calculation of these profiles.

In accord with this suggestion, we modified the manuscript text as follows:

“To define compartments in mosquito genomes, we attempted either to exclude centromeric and telomeric regions from PCA (cropping approach) or computing local PC1 in megabase-scaled frames (framing approach). However, neither cropped nor framed data does not allow capturing of the observed plaid pattern of contacts (Supplementary Note I).”

Reviewer #2 (Remarks to the Author):

In the manuscript, NCOMMS-21-18764-T, Lukyanchikova et al generated Hi-C datasets in five *Anopheles* mosquito species (with embryos or adult mosquito bodies). Firstly, using these datasets, the authors were able to improve existing genome assemblies of three mosquito species and successfully performed de novo assembly for two others, and identified large polymorphic chromosomal rearrangements in these mosquito genomes. Secondly, the Hi-C data revealed the known architectural features of 3D genome organization that have been identified in other species, including the Rabl conformation, A/B compartments, TADs, and Chromatin loops, etc. The authors were able to identify several evolutionally- and developmentally conserved giant chromatin loops that span very large genomic distance (tens of megabases). Lastly, the authors performed evolutionary analysis and found that chromatin architecture within syntenic blocks remains remarkably stable between species.

Major concerns

1. The genome size and ploidy. Are the genomes of these mosquito species diploid or polyploidy?

The genome sizes of *Anopheles* mosquitoes can be found in Table 1 “*Assembly statistics before (reference) and after (de novo) Hi-C assembly*”, columns “*Total scaffold length*” or “*Length of chromosomes*”. Regarding the genome ploidy question - embryos, which are the main experimental material in this study, are largely diploid while many adult tissues are polyploid and contain polytene chromosomes.

2. The authors said that they performed the Hi-C assays in each sample with biological replicates. However, they did not include any result proving the quality of reproducibility.

Thank you for pointing out this omission. Below we provide the Pearson’s correlation coefficient between Hi-C replicates for each species. The data was obtained using distance-normalized contact frequencies at 100 kb resolution. It should be noted that to obtain more Hi-C reads per individual species we performed additional sequencing for some libraries and those cases are defined as technical replicates (tec.rep.) in the Additional Table 1.

Additional Table 1.

	AalbS2_V4 rep 1	AalbS2_V4 rep 2	-	-
AalbS2_V4 rep 1	1.0	0.98	-	-
AalbS2_V4 rep 2	0.98	1.0	-	-
	AatrE3_V4 1	AatrE3_V4 2 (tec.rep 1)	AatrE3_V4 2 (tec.rep 2)	-
AatrE3_V4 rep 1	1.0	0.96	0.97	-
AatrE3_V4 rep 2 (tec.rep 1)	0.96	1.0	0.98	-
AatrE3_V4 rep 2 (tec.rep 2)	0.97	0.98	1.0	-
	AcolNg_V4 rep 1	AcolNg_V4 rep 2	AcolNg_V4 rep 3 (tec.rep 1)	AcolNg_V4 rep 3 (tec.rep 2)
AcolNg_V4 rep 1	1.0	0.94	0.92	0.93
AcolNg_V4 rep 2	0.94	1.0	0.92	0.93
AcolNg_V4 rep 3 (tec.rep 1)	0.92	0.92	1.0	0.95
AcolNg_V4 rep 3 (tec.rep 2)	0.93	0.93	0.95	1.0

	AmerR4A_V4 rep 1	AmerR4A_V4 rep 2	-	-
AmerR4A_V4 rep 1	1.0	0.97	-	-
AmerR4A_V4 rep 2	0.97	1.0	-	-
	AmerR4_V4 rep 1 (tec.rep 1)	AmerR4_V4 rep 1 (tec.rep 2)	AmerR4_V4 rep 2	-
AmerR4_V4 rep 1 (tec.rep 1)	1.0	0.94	0.94	-
AmerR4_V4 rep 1 (tec.rep 2)	0.94	1.0	0.91	-
AmerR4_V4 rep 2	0.94	0.91	1.0	-
	Astel2_V4 rep 1 (tec.rep 1)	Astel2_V4 rep 2	Astel2_V4 rep 1 (tec.rep 2)	Astel2_V4 rep 1 (tec.rep 3)
Astel2_V4 rep 1 (tec.rep 1)	1.0	0.95	0.97	0.97
Astel2_V4 rep 2	0.97	1.0	0.96	0.95
Astel2_V4 rep 1 (tec.rep 2)	0.97	0.96	1.0	0.98
Astel2_V4 rep 1 (tec.rep 3)	0.97	0.95	0.98	1.0

3. A whole embryo or adult body contains different types (different lineages/tissues/organs) of cells and the cell populations are highly heterogeneous. Hence, the observed architectural features of 3D genome organization in each species are averaged result of the different cell

types --- For those features that are largely conserved among different cell lineage/tissue types, such as Rab1 configuration, A/B compartment and TAD organization, this will not be a big problem. However, for the fine-scale features that are more cell type-specific, such as chromatin loops, this will cause problems. A good example for this is the discrepancy regarding the giant loops between Hi-C and the 3D FISH results. From the Hi-C data, the authors said that these long-range loops are both evolutionarily and developmentally conserved, whereas in 3D FISH assays they appeared cell type-specific. To clarify this discrepancy, high-resolution 3C/4C assays in purified cell population might be a good choice.

We agree with Reviewer#2 that the “developmental conservation” of the long-range loops might be too strong to claim in our manuscript. Indeed, as we demonstrated by FISH, at least in some cell types these loops are not present (i.e. in nurse cells with polytene chromosomes with high polyteny level). Moreover, we see variability in the loop formation among individual cells of the same type. What we would like to emphasize is that the formation of loops is not restricted to a single cell type or developmental time point (e.g. embryos), because we observed the loops in both embryos and adult mosquitos. Whereas further studies on cell type-specificity of the loops are interesting, this task is challenging because there are no well-established protocols allowing isolation of pure cell populations from mosquito embryos. The investigation of long loop tissue specificity could become a separate research project, while in the revised manuscript we claimed that “*these loops are not specific to one particular developmental-stage or cell type.*” In addition, we extended the discussion to emphasize the limitations caused by embryo cell heterogeneity (see below the answer to comment #4).

4. Due to the heterogeneity issue described in the above point, the contact frequency decay curves in Fig 10 might not truly reflect what occurs in each cell type in each mosquitoes species.

As follows from Supplementary Figure 27, there are only a few exceptions when chromatin contact scaling varies significantly among cell types. However, we agree that our data may not reflect what occurs in each cell type in each mosquito species. We emphasized this point introducing a new paragraph in the discussion section:

“Finally, we note that all experiments were performed on heterogeneous populations of embryo cells or adult mosquitoes, which include multiple cell types. Thus, we can not discriminate cell type-specific properties of genome architecture; moreover, according to recent studies (Szabo et al. 2020; Ulianov et al. 2021) certain chromatin structures observed at the level of cell populations may not exist in individual cells and, vice versa, chromatin nanodomains of individual cells may not be captured by standard Hi-C approach. Therefore, studying genome organization at the level of individual mosquito cells or pure cell population is an important direction of future research.”

5. In the title the authors claimed Anopheles mosquitoes reveal new principles of 3D genome organization in insects, which seems not quite true, i.e., the data doesn't show much new principles of 3D genome organization in insects.

We would like to highlight new principles revealed by our study:

1) Our work identified at least two types of B compartments in mosquitoes: euchromatic and heterochromatic. Large intercalary heterochromatin blocks, cytogenetically defined as diffuse intercalary heterochromatin (Sharakhova et al. 2010), form B-compartments with regions of pericentromeric heterochromatin, but they do not form B-compartments with euchromatic regions. Euchromatic regions form their own B-compartments, similar to those described in *Drosophila*. Although *D. melanogaster* chromosomes do have compact intercalary heterochromatin, it has a different nature than diffuse intercalary heterochromatin found in *Anopheles*.

2) We uncovered a new class of extremely long, evolutionary conserved chromatin loops in the malaria mosquito genomes, formed by yet unknown mechanisms. This type of loops has not been found either in mammalian or *Drosophila* genomes.

3) Our results largely uncouple the 3D-genome architecture and chromosomal evolution in malaria mosquitoes, which is a novel finding in light of the recent discussion of this subject by the research community (Ghavi-Helm et al. 2019; Renschler et al. 2019; Torosin et al. 2020; Liao et al. 2021); in addition, we showed a confounding effect of gene density on both insulation and distribution of synteny breakpoints. We explained the co-occurrence of topologically associated domain boundaries and synteny breakpoints by the confounder effect of the increased gene density.

Minor

concerns

1. Table S5 meant all the giant loops?

The reviewer's statement is correct, Supplementary Table 5 contains all the giant loops and their genomic coordinates.

2. What are the size of the FISH probes used?

FISH probe sizes varied between 500-1500 bp and were designed for exon regions mostly. This information is now added to the Methods section. We can supply Supplementary Table 11 with column "probe size/length" or "probe start-end position", if the Reviewer requires this information.

3. It seems the giant loops are just structural features without functional implications.

We appreciate your opinion. We can surely claim the giant loops as structural units in the 3D-genomes of Anopheles mosquitoes. To further validate/prove them to be functional units in the genome, we need to perform additional investigations.

Reviewer #3 (Remarks to the Author):

The authors extensively characterize the genomes and genome organizations of five different Anopheles species using Hi-C and other data. The work is an important contribution to the field both as a resource and in terms of the results revealed with respect to the conservation of genome organization across these species and how this relates to other organisms. Also, an additional contribution of the paper is the introduction of a new way to do chromatin compartment analysis (contrast enrichment) when regular PC analysis does not accurately capture active vs inactive compartments, which is generally the case for chromosomes folded in Rab1 configuration across many different organisms. However, the paper can certainly benefit from a bit of streamlining and restructuring to make sure the main points come across. I also have additional comments that need to be addressed before publication.

1. Since the Hi-C-based assemblies they have put together are done through manual annotation, it is imperative to further quality check these using read-level information around

breakpoints to convincingly show that these are assemblies that can be used by others. The authors already indicate some concerns about Hi-C assemblies, as well as PacBio scaffolding, when they mention AgamP4.

Indeed, we manually modified original 3D-DNA assemblies following the best practices of the DNA-zoo genome assemblies project. To show that this manual editing does not decrease assembly quality, we first employed long-read sequencing data which is available for three species (*An. albimanus* (Compton et al. 2020), *An. coluzzi* (Kingan et al. 2019), *An. merus* (our data)). We extracted scaffold breakpoints from assemblies and subdivided them into two categories: 1) breakpoints introduced by 3D-DNA automatic pipeline and 2) breakpoints introduced by manual editing of assemblies using juicebox assembly visualization tool. Next, we aligned long-reads data to the assemblies and computed breakpoints coverage. As follows from the Additional Table 2, in each of the breakpoints categories, a large portion (~30-66%) of breakpoints was covered by at least one read.

Additional Table 2.

Assembly	Number of breakpoints introduced by 3D-DNA / confirmed by long-read sequencing	Number of breakpoints introduced by manual editing of assemblies / confirmed by long-read sequencing
AalbS2_V4	67 / 28	33 / 16
AcolNg_V4	115 / 63	21 / 14
AmerR4_V4	267 / 105	4 / 2

We were not able to validate the remaining breakpoint regions at the reads level using Hi-C data due to the chimeric nature of Hi-C reads. However, we employed properties of genomic contacts distribution to double-check assemblies' quality. We note that contact frequency strongly depends on the genomic distance. Genomic distances are defined by the assembly, whereas scaffolds' contact frequencies do not depend on the scaffolds'

order. Thus, correctly placed scaffolds should display contact frequencies in agreement with their genomic distance, whereas misplacing a scaffold will result in the drop of frequency of contacts between this scaffold and its neighbors. This feature of contact frequencies could be quantified as $\log(\text{observed_cf}/\text{expected_cf})$, where expected_cf is an average contact frequency for all loci pairs at matching genomic distance. Sampling contacts around misassembled scaffolds would shift the $\log(\text{observed_cf}/\text{expected_cf})$ towards negative values.

To check whether manually editing the assembly improved its quality, we extracted genomic distances from two types of assemblies:

- a raw assembly produced by 3D-DNA algorithm
- a modified assembly where some scaffolds were rearranged / split based on visual inspection of Hi-C maps.

We separately sampled contacts around scaffolds that were reshuffled only (without splitting the original scaffold) and scaffolds that were split during manual editing (i.e. a new breakpoint was introduced to the assembly).

The obtained results (Additional Fig. 2) show that the $\log(\text{observed_cf}/\text{expected_cf})$ distribution was shifted toward negative values when using genomic distances from raw assembly produced by 3D-DNA. Importantly, this shift was not observed when comparing contacts of loci within unmodified scaffolds, indicating that the difference between assemblies is due to misplacing of the scaffolds in raw 3D-DNA assembly. Thus, manually edited assembly shows better agreement between genomic distances and contact frequencies.

Additional Figure 2.

Figure legend:

Red solid line - contacts around breakpoints introduced by the manual splitting of the original scaffolds; genomic distances and expected values computed based on the manually-edited assembly

Red dotted line - same regions as above, but genomic distances and expected values computed based on unedited 3D-DNA assembly

Green solid line - contacts around boundaries of scaffold manually placed during assembling; genomic distances and expected values computed based on the manually-edited assembly

Green dotted line - same regions as above, genomic distances and expected values computed based on unedited 3D-DNA assembly

X axis - $\log_2(\text{mean}(\text{observed_cf}/\text{expected_cf}))$, contacts were computed for +/- 50 kb neighborhood of the target locus

Y axis - fraction of loci with corresponding $\log_2(\text{mean}(\text{observed_cf}/\text{expected_cf}))$ value

2. The suggestion that the B compartment has at least two different forms can be followed up using subcompartment analysis as was done before for mammalian genomes.

Following this suggestion, we adapted the method described in Rao et al. 2014 to call the subcompartments in *Anopheles* species. Similar to our results with compartments calling, the subcompartments obtained by standard algorithm reflect a genomic distance of locus to telomere or centromere. We showed several examples of compartment tracks in the Additional Fig. 3 below, and full data could be found on https://genedev.bionet.nsc.ru/site/hic_out/by_Project/Anopheles/ActualData/hmm/. Thus, we concluded that subcompartments could not be identified in *Anopheles* species as was done before for mammalian genomes, and this task requires the development of new algorithms specific for chromosomes with Rabl-configuration.

The figure below shows subcompartments tracks generated for *An. merus*, 3R (4 top tracks), in comparison with A/B-compartments obtained with the cePC1 algorithm and described in the current manuscript. Different subcompartments are shown in different colors. Following the algorithm from Rao et al., 2014, we used one chromosomal element to define interchromosomal contacts for subcompartments calling; the name of the chromosome used for track generation is provided in the track label.

Additional Figure 3. Different subcompartments identified for *An.merus* using the standard algorithm from Rao et al., 2014.

A.

B.

3. The authors should discuss the potential relationship of the giant loops that are located outside of H3K27me3-rich regions to DNA methylation nadirs described by Goodell lab in 2020.

Thank you for pointing to that great manuscript, which was missing in our discussion. Indeed, interactions between methylation grand canyons also result in formation of extremely long-range loops. However, we note that these loops are associated with H3K27me3 signal (“we found that nearly all grand canyons (85% [241 of 282]) exhibit broad H3K27me3”, Zhang et al. 2020), which is not the case for the A-loops identified in

Anopheles

Hi-C

maps.

We have discussed DNA methylation nadirs in comparison with *Anopheles* loops in the paragraph “*Evolutionarily conserved long-range chromatin interactions in Anopheles genomes*” section in the manuscript:

“The second type, represented by giant X-chromosome loops, shows moderate enrichment of H3K27me3 signal and, thus, could be also associated with Polycomb complexes, similar to the long-distance loops described previously in mammals (Kraft et al. 2020; Zhang et al. 2020).”

and

“The absence of H3K27me3-signal makes these loops different from long-range loops observed previously (Kraft et al. 2020; Zhang et al. 2020).”

4. They need to elaborate on "confirmed that all these cases were due to errors in TAD or cePC1 calling". Were these mainly issues in compartment or TAD calling? I suspect mainly due to TAD calls. They need to show aggregate plots around such problematic calls to show that cePC1 values do not behave similarly when they are around non-problematic boundaries.

Following this advice, we separately plotted cePC1 values around “problematic” and “normal” TAD boundaries. It should be noted that we selected the problematic regions as “cases where a strong TAD boundary (top quartile of insulation score distribution) separates regions with similar cePC1 values (cePC1 difference belongs to the bottom quartile of the distribution)”. As follows from the definition of normal and problematic TADs, differences of the cePC1 values across normal boundaries were much more pronounced. Since aggregate plots look similar for all five species, we provided results for *An. albimanus* (Additional Fig. 4).

Additional Figure 4. cePC1 values across TAD boundaries.

A. Normal boundaries

B. “Problematic” boundaries

We next aimed to examine in detail the “problematic” boundaries. There are ~35-45 such boundaries in each species. We classified them manually into three categories:

1. We observe TAD boundary, and in the same location we observe a switch of cePC1 values sign (i.e. A-B compartment switch), but there is 1-2 bins discrepancy between cePC1 and TAD boundary manifestation. We assume that this is because TADs and cePC1 couldn't be called at the same resolution (TADs are called at 5-kb, whereas cePC1 at 25-kb resolution)
2. cePC1 values are not defined correctly, i.e. we can visually see plaid-pattern, and this pattern explains a TAD boundary, but cePC1 values indicate a single contiguous compartment. This often happens when compartments are relatively small and long-range interactions are noisy.

Examining chromosome 2L of *An. albimanus* (14 problematic TAD boundaries out of 35 identified in this species) we found five examples belonging to the first category, and nine examples belonging to the second category. We provided representative images below, at the Additional Fig.5.

Additional Figure 5.

A. cePC1 change of value and TAD boundary are located within 1-2 bins from each other

B. cePC1 values are not defined correctly

Thus, we concluded that there might indeed be several cases of local insulation inconsistent with the long-range interactions pattern, but the overwhelming majority of TAD boundaries are concordant with cePC1-values and genomic compartments. To reflect this conclusion, we modified the article text as follows:

“We additionally inspected all cases where a strong TAD boundary (top quartile of insulation score distribution) separated regions with similar cePC1 values (cePC1 difference belongs to the bottom quartile of the distribution). Some of these cases

appeared to be due to different resolutions of cePC1 vector (computed at 25-kb binned matrix) and TAD-boundaries (called at 5-kb resolutions). We also found a limited number of wrongly defined cePC1 calls.”

5. For X-loop and A-loop, and for other similar extremely long ones, sufficiently detailed analysis at the read-level is needed to eliminate the possibility that they correspond to structural variation at the DNA level rather than loops. I am not a FISH expert; therefore, I cannot be sure whether the differences between low and high polytene chromosomes rule out this possibility.

In addition to quite convincing FISH-experiments mentioned above, we have now obtained evidence from long-read data to ensure that long-range loops are not assembly artifacts. We aligned PacBio long reads available for *An. coluzzii* and *An. merus* (Kingan et al. 2019; our data), and Oxford Nanopore reads - for *An. albimanus* (Compton et al. 2020) to the assembled genomes and found no chimeric reads, containing two anchors of a loop (the chimeric reads track is available at https://genedev.bionet.nsc.ru/site/hic_out/by_Species/Invertebrates/Anopheles/ActualData/longreads/). This suggests that despite the observed interaction of loop anchors in 3D-space, these anchors are not neighbors in the 1D-genome sequences. Next, for each loop anchor, we were able to find overlapping long-read alignments covering the anchor and its neighboring scaffold (exemplified in the Additional Fig.6, below). This ensures that loops are correctly placed in our assembly.

Additional Figure 6.

A. *An.coluzzii*_X-loop

B. *An.coluzzii*_2R-loop

Additional Figure 6 (continued).

*C. An.merus*_X-loop

*D. An.merus*_2R-loop

Additional Figure 6 (continued).

E. *An.albimanus*_X-loop

F. *An.albimanus*_2R-loop

In addition, we note that the loops have syntenic anchors, and misplacing all these syntenic scaffolds in all five independently assembled genomes is unlikely.

6. Extremely long-distance loops are presented and listed in supplementary; however, other loops have not been characterized in much detail beyond the H3K27me3 association.

Except for long-distance loops, we noticed other chromatin loops in *Anopheles* genomes and briefly mentioned that fact in “*Hi-C maps identified long-range interactions and FISH validated chromatin loops*” paragraph. Their anchors lay much closer than in case of A- and X-loops, and based on H3K27me3 ChIP-seq results were associated with Polycomb protein complex. That type of chromatin loop was previously described in detail in *Drosophila* genomes in several publications (Sexton et al. 2012; Eagen et al. 2017). Because the main focus of our research was on long-range loops uniquely observed in *Anopheles* mosquitoes, we decided to characterize mostly them but not the other loops. However, the set of Polycomb loops specific for *An. atroparvus* and their genomic coordinates can be found in Supplementary Table 10.

Additionally, following this reviewer's suggestion, we have called chromatin loops in five *Anopheles* species using the algorithm described in (Rao et al. 2014). All produced loop files can be found at https://genedev.bionet.nsc.ru/site/hic_out/by_Project/Anopheles/ActualData/loops/. We found that some of detected loops correspond correctly to the bright dots visible on the Hi-C map, overlap with H3K27me3-enriched regions, and, indeed, reflect the actual chromatin interactions (Additional Fig. 7, A-C). However, the majority of automatically annotated loops are wrongly placed (Additional Fig. 7, D-I) or missing (Additional Fig. 7, C). Based on examples shown in Fig. 7, we speculated that large blocks of highly insulated pericentromeric heterochromatin impede the loop calling using the standard algorithms and decrease the biological value of detected ones.

Additional Figure 7.

- For each Hi-C heatmap, the authors need to indicate the resolution used (10kb? 25kb?) and a color scale.

We have indicated the resolution and color bar for each Hi-C heatmap on the main and supplementary figures.

- conservative features -> conserved features

Thank you for pointing to the typo. We fixed it in the revised version of the manuscript.

- H3K27me3-reach -> H3K27me3-rich

Thank you for pointing to the typo. We fixed it in the revised version of the manuscript.

- "active A-compartment (gene dense, expressed, GC-rich)": as their correlation tables in the supplement indicate, for most of the species, the cePC1 values did not correlate with GC content. Therefore the sentence containing this part needs to reflect the data.

Thank you for this suggestion. We modified this sentence as follows:

“We developed a computational approach to detect compartments and showed that spatial compartmentalization distinguishes the active A-compartment (gene dense, actively expressed) and inactive B-compartment (gene-poor, mostly silent).”

References:

- Compton A, Liang J, Chen C, Lukyanchikova V, Qi Y, Potters M, Settlage R, Miller D, Deschamps S, Mao C, et al. 2020. The Beginning of the End: A Chromosomal Assembly of the New World Malaria Mosquito Ends with a Novel Telomere. *G3* 10:3811–3819.
- Eagen KP, Aiden EL, Kornberg RD. 2017. Polycomb-mediated chromatin loops revealed by a subkilobase-resolution chromatin interaction map. *Proc Natl Acad Sci USA* 114:8764–8769.
- Ghavi-Helm Y, Jankowski A, Meiers S, Viales RR, Korbelt JO, Furlong EEM. 2019. Highly rearranged chromosomes reveal uncoupling between genome topology and gene expression. *Nat Genet* 51:1272–1282.
- Kaushal A, Mohana G, Dorier J, Özdemir I, Omer A, Cousin P, Semenova A, Taschner M, Dergai O, Marzetta F, et al. 2021. CTCF loss has limited effects on global genome architecture in *Drosophila* despite critical regulatory functions. *Nat Commun* 12:1011.
- Kingan S, Heaton H, Cudini J, Lambert C, Baybayan P, Galvin B, Durbin R, Korlach J, Lawniczak M. 2019. A High-Quality De novo Genome Assembly from a Single Mosquito Using PacBio Sequencing. *Genes* 10:62.
- Kraft K, Yost KE, Murphy S, Magg A, Long Y, Corces MR, Granja JM, Mundlos S, Cech TR, Boettiger A, et al. 2020. Polycomb-mediated Genome Architecture Enables Long-range Spreading of H3K27 methylation. Genetics Available from: <http://biorxiv.org/lookup/doi/10.1101/2020.07.27.223438>
- Liao Y, Zhang X, Chakraborty M, Emerson JJ. 2021. Topologically associating domains and their role in the evolution of genome structure and function in *Drosophila*. *Genome Res.* 31:397–410.
- Loubiere V, Delest A, Thomas A, Bonev B, Schuettengruber B, Sati S, Martinez A-M, Cavalli G. 2016. Coordinate redeployment of PRC1 proteins suppresses tumor formation during *Drosophila* development. *Nat Genet* 48:1436–1442.
- Loubiere V, Papadopoulos GL, Szabo Q, Martinez A-M, Cavalli G. 2020. Widespread activation of developmental gene expression characterized by PRC1-dependent chromatin looping. *Sci. Adv.* 6:eaax4001.
- Rao SSP, Huntley MH, Durand NC, Stamenova EK, Bochkov ID, Robinson JT, Sanborn AL, Machol I, Omer AD, Lander ES, et al. 2014. A 3D Map of the Human Genome at Kilobase Resolution Reveals Principles of Chromatin Looping. *Cell* 159:1665–1680.
- Renschler G, Richard G, Valsecchi CIK, Toscano S, Arrigoni L, Ramírez F, Akhtar A. 2019. Hi-C guided assemblies reveal conserved regulatory topologies on X and autosomes despite extensive genome shuffling. *Genes Dev.* 33:1591–1612.
- Rowley MJ, Nichols MH, Lyu X, Ando-Kuri M, Rivera ISM, Hermetz K, Wang P, Ruan Y, Corces VG. 2017. Evolutionarily Conserved Principles Predict 3D Chromatin Organization. *Molecular Cell* 67:837-852.e7.
- Schuettengruber B, Ganapathi M, Leblanc B, Portoso M, Jaschek R, Tolhuis B, van Lohuizen M, Tanay A, Cavalli G. 2009. Functional Anatomy of Polycomb and Trithorax Chromatin Landscapes in *Drosophila* Embryos. Kingston R, editor. *PLoS Biol* 7:e1000013.
- Sexton T, Yaffe E, Kenigsberg E, Bantignies F, Leblanc B, Hoichman M, Parrinello H, Tanay A, Cavalli G. 2012. Three-Dimensional Folding and Functional Organization Principles of the *Drosophila* Genome. *Cell* 148:458–472.

- Sharakhova MV, George P, Brusentsova IV, Leman SC, Bailey JA, Smith CD, Sharakhov IV. 2010. Genome mapping and characterization of the *Anopheles gambiae* heterochromatin. *BMC Genomics* 11:459.
- Szabo Q, Donjon A, Jerković I, Papadopoulos GL, Cheutin T, Bonev B, Nora EP, Bruneau BG, Bantignies F, Cavalli G. 2020. Regulation of single-cell genome organization into TADs and chromatin nanodomains. *Nat Genet* 52:1151–1157.
- Torosin NS, Anand A, Golla TR, Cao W, Ellison CE. 2020. 3D genome evolution and reorganization in the *Drosophila melanogaster* species group. Payseur B, editor. *PLoS Genet* 16:e1009229.
- Ulianov SV, Zakharova VV, Galitsyna AA, Kos PI, Polovnikov KE, Flyamer IM, Mikhaleva EA, Khrameeva EE, Germini D, Logacheva MD, et al. 2021. Order and stochasticity in the folding of individual *Drosophila* genomes. *Nat Commun* 12:41.
- Zhang X, Jeong M, Huang X, Wang XQ, Wang X, Zhou W, Shamim MS, Gore H, Himadewi P, Liu Y, et al. 2020. Large DNA Methylation Nadirs Anchor Chromatin Loops Maintaining Hematopoietic Stem Cell Identity. *Molecular Cell* 78:506-521.e6.

Reviewers' Comments:

Reviewer #1:

Remarks to the Author:

The authors have properly addressed all my critical remarks. The MS may be published in the current form

Reviewer #2:

Remarks to the Author:

In the revised manuscript, the authors included some new data and clarified the concerns raised by the reviewers in the 1st-round of review. This reviewer is satisfied with the improvements of the manuscript and support its publication in the journal of Nature Communications.

Reviewer #3:

Remarks to the Author:

The authors have done a great job in addressing all my major concerns. I only have one minor comment remaining below but am supportive of this paper's acceptance regardless.

Previous Comment 1: The additional analysis with spanning long reads and expected contact patterns from Hi-C around the breakpoints certainly supports their assemblies. However, for Hi-C, I would have liked plots that are easier to interpret such as aggregate Hi-C heatmaps around the breakpoints or genomic distance scaling plots from the bins flanking the breakpoint region.

Rebuttal letter R2

We are deeply grateful to the Reviewers for taking their time to review the updated manuscript. Below, we provide our responses (in bold text) point by point to each comment.

REVIEWER COMMENTS

Reviewer #1 (Remarks to the Author):

The authors have properly addressed all my critical remarks. The MS may be published in the current form

We thank the reviewer for positive assessment of the updated manuscript.

Reviewer #2 (Remarks to the Author):

In the revised manuscript, the authors included some new data and clarified the concerns raised by the reviewers in the 1st-round of review. This reviewer is satisfied with the improvements of the manuscript and support its publication in the journal of Nature Communications.

We thank the reviewer for positive assessment of the updated manuscript.

Reviewer #3 (Remarks to the Author):

The authors have done a great job in addressing all my major concerns. I only have one minor comment remaining below but am supportive of this paper's acceptance regardless.

Previous Comment 1: The additional analysis with spanning long reads and expected contact patterns from Hi-C around the breakpoints certainly supports their assemblies. However, for Hi-C, I would have liked plots that are easier to interpret such as

aggregate Hi-C heatmaps around the breakpoints or genomic distance scaling plots from the bins flanking the breakpoint region.

As per your advice, we provided aggregated scaling plots (figure 1 below). For this aim, we:

- 1) selected all genomic bins located 100-kb away from breakpoint;**
 - 2) computed aggregated genomic distance scaling plots for selected bins using either raw or manually modified assembly, where some scaffolds were rearranged / split based on visual inspection of Hi-C maps.**
- As previously, we separately provided data for breakpoints where scaffolds were reshuffled only (without splitting the original scaffold; green line) and scaffolds that were split during manual editing (i.e. a new breakpoint was introduced to the assembly; red line).**

For manually modified assemblies (solid line), contact frequency of the bins flanking the breakpoint region does not differ from genome-wide average (i.e. $\log(\text{observed/expected}) \sim 0$ and contacts scaling does not change after breakpoint). However, for *An. albimanus*, *An. atroparvus*, and *An. stephensi* raw assemblies, frequencies of contacts between bins separated by a breakpoint (distances exceeding 100 kb) are less than genome-wide average. This indicates that placement of these scaffolds is not concordant with Hi-C maps, and suggests that manual assemblies have higher quality.

Figure 1.